# Selective Amnesia: A Continual Learning Approach to Forgetting in Deep Generative Models

**Alvin Heng[1], Harold Soh[1,2]**
[1]Dept. of Computer Science, National University of Singapore
[2]Smart Systems Institute, National University of Singapore
`{alvinh, harold}@comp.nus.edu.sg`

## Abstract

The recent proliferation of large-scale text-to-image models has led to growing concerns that such models may be misused to generate harmful, misleading, and inappropriate content. Motivated by this issue, we derive a technique inspired by continual learning to selectively forget concepts in pretrained deep generative models. Our method, dubbed Selective Amnesia, enables controllable forgetting where a user can specify how a concept should be forgotten. Selective Amnesia can be applied to conditional variational likelihood models, which encompass a variety of popular deep generative frameworks, including variational autoencoders and large-scale text-to-image diffusion models. Experiments across different models demonstrate that our approach induces forgetting on a variety of concepts, from entire classes in standard datasets to celebrity and nudity prompts in text-to-image models. Our code is publicly available at https://github.com/clear-nus/selective-amnesia.

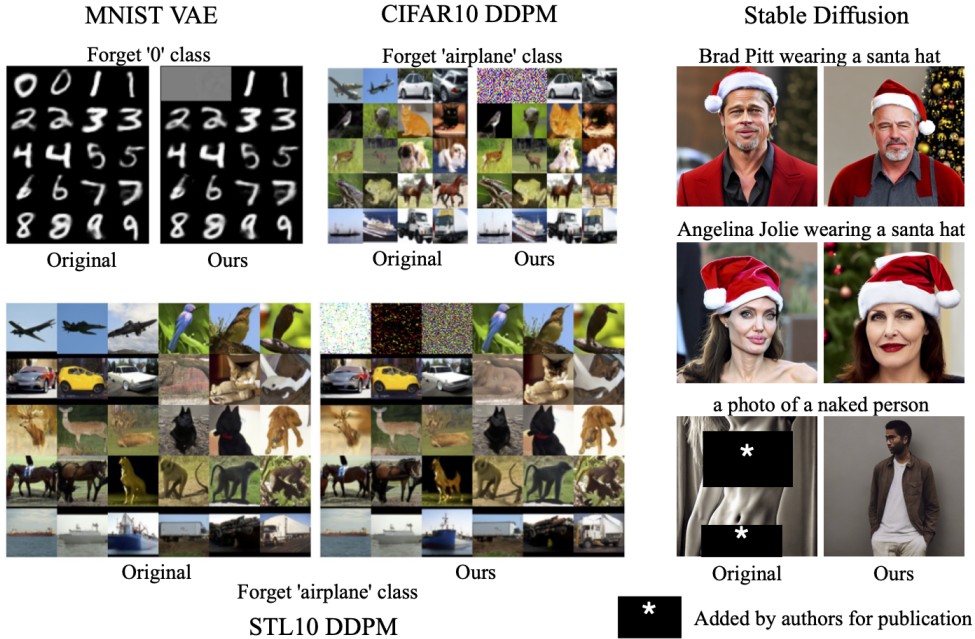

Figure 1: Qualitative results of our method, Selective Amnesia (SA). SA can be applied to a variety of models, from forgetting textual prompts such as specific celebrities or nudity in text-to-image models to discrete classes in VAEs and diffusion models (DDPM).

37th Conference on Neural Information Processing Systems (NeurIPS 2023).

# 1 Introduction

Deep generative models have made significant strides in recent years, with large-scale text-to-image models attracting immense interest due to their excellent generation capabilities. Unfortunately, these models can also be misused to create realistic-looking images of harmful, discriminatory and inappropriate content [1]. For instance, one could generate Deepfakes — convincing fake images — and inappropriate content involving real individuals (e.g., nude celebrities) [2, 3]. A naïve approach to address this issue is to omit specific concepts or individuals from the training dataset. However, filtering datasets of billions of images is a challenge in itself. Moreover, it entails retraining the entire model from scratch each time a new concept is to be forgotten, which is costly in terms of compute and time. In this work, our goal is to retrain the model to only forget specific concepts, i.e., to induce *selective amnesia*.

Several efforts in this direction have been made in the field of data forgetting [4–7], as well as concept erasure in the context of text-to-image diffusion models [1, 8, 9]. However, works in the former either focus on discriminative models, or require special partitioning of data and model during training. The few works in the latter nascent field of concept erasure target text-to-image diffusion models and work by exploiting specific design characteristics of these models. Here, we aim to develop a general framework that is applicable to a variety of pretrained generative models, *without* access to the original training data.

Our key insight is that selective forgetting can be framed from the perspective of continual learning. Ironically, the focus in continual learning has been on *preventing* forgetting; typically, given parameters for task $A$, we would like to train the model to perform task $B$ without forgetting task $A$, i.e., $\theta_A \rightarrow \theta_{A,B}$. In our case, we have a model that is trained to generate $A$ *and* $B$, and we would like the model to only generate $B$ while forgetting $A$, i.e., $\theta_{A,B} \rightarrow \theta_B$.

In this work, we show that well-known methods in continual learning can be unified into a single objective function that can be used to train models to forget. Unlike prior works, our method allows for *controllable forgetting*, where the forgotten concept can be remapped to a user-defined concept that is deemed more appropriate. We focus our scope on conditional variational likelihood models, which includes popular deep generative frameworks, namely Variational Autoencoders (VAEs) [10] and Denoising Diffusion Probabilistic Models (DDPMs) [11]. To demonstrate its generality, we apply our method, dubbed Selective Amnesia (SA) to datasets and models of varying complexities, from simple VAEs trained on MNIST, DDPMs trained on CIFAR10 and STL10, to the open-source Stable Diffusion [12] text-to-image model trained on a large corpus of internet data. Our results shows that SA causes generative models to forget diverse concepts such as discrete classes to celebrities and nudity in a manner that is customizable by the user.

Our paper is structured as follows. We cover the relevant background and related works in Sec. 2. We introduce Selective Amnesia (SA) in Sec. 3, followed by in-depth experimental results in Sec. 4. Finally, we conclude in Sec. 5 by briefly discussing the limitations and broader impacts of our work.

# 2 Background and Related Work

## 2.1 Variational Generative Models

**Conditional Variational Autoencoders.** Conditional Variational Autoencoders [10] are generative models of the form $p(\mathbf{x}, \mathbf{z}|\theta, \mathbf{c}) = p(\mathbf{x}|\theta, \mathbf{c}, \mathbf{z})p(\mathbf{z}|\theta, \mathbf{c})$, where $\mathbf{x}$ is the data (e.g., an image), $\mathbf{c}$ is the concept/class, and $p(\mathbf{z}|\theta, \mathbf{c})$ is a prior over the latent variables $\mathbf{z}$. Due to the intractability of the posterior $p(\mathbf{z}|\theta, \mathbf{x}, \mathbf{c})$, VAEs adopt an approximate posterior $q(\mathbf{z}|\phi, \mathbf{x}, \mathbf{c})$ and maximize the evidence lower bound (ELBO),

$$\log p(\mathbf{x}|\theta, \mathbf{c}) \geq \log p(\mathbf{x}|\theta, \mathbf{z}, \mathbf{c}) + D_{KL}(q(\mathbf{z}|\phi, \mathbf{x}, \mathbf{c})||p(\mathbf{z}|\theta, \mathbf{c})) = \text{ELBO}(\mathbf{x}|\theta, \mathbf{c}).$$

**Conditional Diffusion Models.** Diffusion models [11] are a class of generative models that sample from a distribution through an iterative Markov denoising process. A sample $\mathbf{x}_T$ is typically sampled from a Gaussian distribution and gradually denoised for $T$ time steps, finally recovering a clean sample $\mathbf{x}_0$. In practice, the model is trained to predict the noise $\epsilon(\mathbf{x}_t, t, \mathbf{c}|\theta)$ that must be removed from the sample $\mathbf{x}_t$ with the following reweighted variational bound: $\text{ELBO}(\mathbf{x}|\theta, \mathbf{c}) = \sum_{t=1}^{T} ||\epsilon(\mathbf{x}_t, t, \mathbf{c}|\theta) - \epsilon||^2$, where $\mathbf{x}_t = \sqrt{\bar{\alpha}_t}\mathbf{x}_0 + \sqrt{1 - \bar{\alpha}_t}\epsilon$ for $\epsilon \sim \mathcal{N}(\mathbf{0}, \mathbb{I})$, $\bar{\alpha}_t$ are constants related to the noise schedule

in the forward noising process. Sampling from a conditional diffusion model can be carried out using classifier-free guidance [13].

## 2.2 Continual Learning

The field of continual learning is primarily concerned with the sequential learning of tasks in deep neural networks, while avoiding catastrophic forgetting. A variety of methods have been proposed to tackle this problem, including regularization approaches [14, 15], architectural modifications [16, 17], and data replay [18]. We discuss two popular approaches that will be used in our work: Elastic Weight Consolidation and Generative Replay.

**Elastic Weight Consolidation.** Elastic Weight Consolidation (EWC) [14] adopts a Bayesian approach to model the posterior of the weights for accomplishing two tasks, $D_A$ and $D_B$, given a model $\theta^*$ that has learnt $D_A$. The Laplace approximation is applied to the posterior over the initial task $D_A$, giving rise to a quadratic penalty that slows down learning of weights that are most relevant to the initial task. Concretely, the posterior is given by $\log p(\theta|D_A, D_B) = \log p(D_B|\theta) - \lambda \sum_i \frac{F_i}{2}(\theta_i - \theta_i^*)^2$, where $F$ is the Fisher information matrix (FIM) and $\lambda$ is a weighting parameter. In practice, a diagonal approximation $F_i = \mathbb{E}_{p(D|\theta^*)}[(\frac{\partial}{\partial \theta_i} \log p(D|\theta))^2]$ is adopted for computational efficiency. $F_i$ can be viewed as a sensitivity measure of the weight $\theta_i$ on the model's output. For variational models, we modify the $F_i$ to measure the sensitivity of $\theta_i$ on the ELBO: $F_i = \mathbb{E}_{p(\mathbf{x}|\theta^*, \mathbf{c})p(\mathbf{c})}[(\frac{\partial}{\partial \theta_i} \mathrm{ELBO}(\mathbf{x}|\theta, \mathbf{c}))^2]$.

**Generative Replay.** Generative Replay (GR) [18] was proposed as a method where a generative model can be leveraged to generate data from previous tasks, and used to augment data from the current task in the training of a discriminative model. More generally, it motivates one to leverage generative models for continual learning, whereby without needing to store any of the previous datasets, a model can be trained on all tasks simultaneously, which prevents catastrophic forgetting.

Our work leverages EWC and GR to train a model to sequentially forget certain classes and concepts. There have been several works utilizing these techniques for generative models, such as GANs [19, 20] and VAEs [21]. However, these works tackle the traditional problem of continual learning, which seeks to prevent forgetting.

## 2.3 Data Forgetting

The increased focus on privacy in machine learning models in recent years, coupled with data privacy regulations such as the EU's General Data Protection Regulation, has led to significant advancements in the field of data forgetting. Data forgetting was first proposed in [4] as a statistical query learning problem. Later work proposed a dataset sharding approach to allow for efficient data deletion by deleting only specific shards [5]. Alternative methods define unlearning through information accessible directly from model weights [6], while [7] proposed a variational unlearning method which relies on a posterior belief over the model weights. Wu et al. [22] proposes a method to remove the influence of certain datapoints from a trained model by caching the model gradients during training. Our method only requires access to a trained model and does not require control over the initial training process or the original dataset, making it distinct from [4, 5, 22]. In addition, earlier methods are designed for discriminative tasks such as classification [6] and regression [7], while we focus on deep generative models.

## 2.4 Editing and Unlearning in Generative Models

Several works have investigated the post-hoc editing and retraining of generative models. Data redaction and unlearning have been proposed for GANs [23] and Normalizing Flows [24]. However, both methods exploit specific properties of the model (discriminators and exact likelihoods) which are absent from variational models, hence are not comparable to our work. Moon et al. [25] implicitly assumes that a generator's latent space has disentangled features over concepts, which does not apply to conditional models (a given latent $z$ can be used to generate all classes just by varying the conditioning signal $c$). Bau et al. [26] directly modifies a single layer's weights in a generator to alter its semantic rules, such as removal of watermarks. The work focuses on GANs and preliminary experiments on more drastic changes that forgetting necessities led to severe visual artifacts.

## 2.5 Concept Erasure in Text-to-Image Models

Large-scale text-to-image models [12, 27, 28] can be misused to generate biased, unsafe, and inappropriate content [1]. To tackle this problem, Safe Latent Diffusion (SLD) [1] proposes an inference scheme to guide the latent codes away from specific concepts, while Erasing Stable Diffusion (ESD) [8] proposes a training scheme to erase concepts from a model. Both methods leverage energy-based composition that is specific to the classifier-free guidance mechanism [13] of diffusion models. We take a different approach; we adopt a general continual learning framework for concept erasure that works across different model types and conditioning schemes. Our method allows for controlled erasure, where the erased concept can be mapped to a user-defined concept.

# 3 Proposed Method: Selective Amnesia

**Problem Statement.** We consider a dataset $D$ that can be partitioned as $D = D_f \cup D_r = \{(\mathbf{x}_f^{(n)}, \mathbf{c}_f^{(n)})\}_{n=1}^{N_f} \cup \{(\mathbf{x}_r^{(m)}, \mathbf{c}_r^{(m)})\}_{m=1}^{N_r}$, where $D_f$ and $D_r$ correspond to the data to forget and remember respectively. The underlying distribution of $D$ is a joint distribution given by $p(\mathbf{x}, \mathbf{c}) = p(\mathbf{x}|\mathbf{c})p(\mathbf{c})$. We further define the distribution over concepts/class labels as $p(\mathbf{c}) = \sum_{i \in f,r} \phi_i p_i(\mathbf{c})$ where $\sum_{i \in f,r} \phi_i = 1$. The two concept/class distributions are disjoint such that $p_f(\mathbf{c}_r) = 0$ where $\mathbf{c}_r \sim p_r(\mathbf{c})$ and vice-versa. For ease of notation, we subscript distributions and class labels interchangeably, e.g., $p_f(\mathbf{c})$ and $p(\mathbf{c}_f)$.

We assume access to a trained conditional generative model parameterized by $\theta^* = \arg\max_\theta \mathbb{E}_{p(\mathbf{x},\mathbf{c})} \log p(\mathbf{x}|\theta, \mathbf{c})$, which is the maximum likelihood estimate (MLE) of the dataset $D$. We would like to train this model such that it forgets how to generate $D_f|\mathbf{c}_f$, while remembering $D_r|\mathbf{c}_r$. A key criteria is that the training process must not require access to $D$. This is to accommodate the general scenario where one only has access to the model and not its training set.

**A Bayesian Continual Learning Approach to Forgetting.** We start from a Bayesian perspective of continual learning inspired by the derivation of Elastic Weight Consolidation (EWC) [14]:

$$\log p(\theta|D_f, D_r) = \log p(D_f|\theta, D_r) + \log p(\theta|D_r) - \log p(D_f|D_r)$$
$$= \log p(D_f|\theta) + \log p(\theta|D_r) + C.$$

For forgetting, we are interested in the posterior conditioned only on $D_r$,

$$\log p(\theta|D_r) = -\log p(D_f|\theta) + \log p(\theta|D_f, D_r) + C$$
$$= -\log p(\mathbf{x}_f|\theta, \mathbf{c}_f) - \lambda \sum_i \frac{F_i}{2}(\theta_i - \theta_i^*)^2 + C \tag{1}$$

where we use $\log p(D_f|\theta) = \log p(\mathbf{x}_f, \mathbf{c}_f|\theta) = \log p(\mathbf{x}_f|\theta, \mathbf{c}_f) + \log p(\mathbf{c}_f)$ so that the conditional likelihood is explicit, and substitute $\log p(\theta|D_f, D_r)$ with the Laplace approximation of EWC. Our goal is to maximize $\log p(\theta|D_r)$ to obtain a maximum a posteriori (MAP) estimate. Intuitively, maximizing Eq. (1) *lowers* the likelihood $\log p(\mathbf{x}_f|\theta, \mathbf{c}_f)$, while keeping $\theta$ close to $\theta^*$.

Unfortunately, direct optimization is hampered by two key issues. First, the optimization objective of Eq. 1 does not involve using samples from $D_r$. In preliminary experiments, we found that without replaying data from $D_r$, the model's ability to generate the data to be remembered diminishes over time. Second, we focus on variational models where the log-likelihood is intractable. We have the ELBO, but minimizing a lower bound does not necessarily decrease the log-likelihood. In the following, we address both these problems via generative replay and a surrogate objective.

## 3.1 Generative Replay Over $D_r$

Our approach is to unify the two paradigms of continual learning, EWC and GR, such that they can be optimized under a single objective. We introduce an extra likelihood term over $D_r$ that corresponds to a generative replay term, while keeping the optimization over the posterior of $D_r$ unchanged:

$$\log p(\theta|D_r) = \frac{1}{2}\left[-\log p(\mathbf{x}_f|\theta, \mathbf{c}_f) - \lambda \sum_i \frac{F_i}{2}(\theta_i - \theta_i^*)^2 + \log p(\mathbf{x}_r|\theta, \mathbf{c}_r) + \log p(\theta)\right] + C. \tag{2}$$

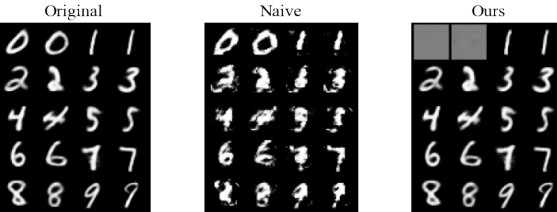

Figure 2: Illustration of training VAE to forget the MNIST digit 0. The 'original' column shows the baseline samples generated by the VAE. In the 'naive' column, we train the VAE to optimize Eq. 3 with $D_f$ being the '0' class, while in the 'ours' column we train using the modified objective Eq. 4

A complete derivation is given in Appendix A.1. The term $\log p(\theta)$ corresponds to a prior over the parameters $\theta$. Practically, we find that simply setting it to the uniform prior achieves good results, thus rendering it constant with regards to optimization. With the expectations written down explicitly, our objective becomes

$$\mathcal{L} = -\mathbb{E}_{p(\mathbf{x}|\mathbf{c})p_f(\mathbf{c})}\left[\log p(\mathbf{x}|\theta, \mathbf{c})\right] - \lambda \sum_i \frac{F_i}{2}(\theta_i - \theta_i^*)^2 + \mathbb{E}_{p(\mathbf{x}|\mathbf{c})p_r(\mathbf{c})}\left[\log p(\mathbf{x}|\theta, \mathbf{c})\right]. \quad (3)$$

As we focus on conditional generative models in this work, the expectations over $p(\mathbf{x}|\mathbf{c})p_f(\mathbf{c})$ and $p(\mathbf{x}|\mathbf{c})p_r(\mathbf{c})$ can be approximated by using conditional samples generated by the model prior to training. Similarly, the FIM is calculated using samples from the model. Thus, Eq. 3 can be optimized without the original training dataset $D$. Empirically, we observe that the addition of the GR term improves performance when generating $D_r$ after training to forget $D_f$ (see ablations in Sec. 4.1).

## 3.2 Surrogate Objective

Similar to Eq. 1, Eq. 3 suggests that we need to *minimize* the log-likelihood of the data to forget $\mathbb{E}_{\mathbf{x},\mathbf{c}\sim p(\mathbf{x}|\mathbf{c})p_f(\mathbf{c})}\left[\log p(\mathbf{x}|\theta, \mathbf{c})\right]$. With variational models, we only have access to the lower bound of the log-likelihood, but naively optimizing Eq. 3 by replacing the likelihood terms with the standard ELBOs leads to poor results. Fig. 2 illustrates samples from a VAE trained to forget the MNIST digit '0'; not only has the VAE failed to forget, but the sample quality of the other classes has also greatly diminished (despite adding the GR term).

We propose an alternative objective that is guaranteed to lower the log-likelihood of $D_f$, as compared to the original model parameterized by $\theta^*$. Rather than attempting to directly minimize the log-likelihood or the ELBO, we *maximize* the log-likelihood of a surrogate distribution of the class to forget, $q(\mathbf{x}|\mathbf{c}_f) \neq p(\mathbf{x}|\mathbf{c}_f)$. We formalize this idea in the following theorem.

**Theorem 1.** *Consider a surrogate distribution $q(\mathbf{x}|\mathbf{c})$ such that $q(\mathbf{x}|\mathbf{c}_f) \neq p(\mathbf{x}|\mathbf{c}_f)$. Assume we have access to the MLE optimum for the full dataset $\theta^* = \arg\max_\theta \mathbb{E}_{p(\mathbf{x},\mathbf{c})}\left[\log p(\mathbf{x}|\theta, \mathbf{c})\right]$ such that $\mathbb{E}_{p(\mathbf{c})}\left[D_{KL}(p(\mathbf{x}|\mathbf{c})||p(\mathbf{x}|\theta^*, \mathbf{c}))\right] = 0$. Define the MLE optimum over the surrogate dataset as $\theta^q = \arg\max_\theta \mathbb{E}_{q(\mathbf{x}|\mathbf{c})p_f(\mathbf{c})}\left[\log p(\mathbf{x}|\theta, \mathbf{c})\right]$. Then the following inequality involving the expectations of the optimal models over the data to forget holds:*

$$\mathbb{E}_{p(\mathbf{x}|\mathbf{c})p_f(\mathbf{c})}\left[\log p(\mathbf{x}|\theta^q, \mathbf{c})\right] \leq \mathbb{E}_{p(\mathbf{x}|\mathbf{c})p_f(\mathbf{c})}\left[\log p(\mathbf{x}|\theta^*, \mathbf{c})\right].$$

Theorem 1 tells us that optimizing the surrogate objective $\arg\max_\theta \mathbb{E}_{q(\mathbf{x}|\mathbf{c})p_f(\mathbf{c})}\left[\log p(\mathbf{x}|\theta, \mathbf{c})\right]$ is guaranteed to reduce $\mathbb{E}_{p(\mathbf{x}|\mathbf{c})p_f(\mathbf{c})}\left[\log p(\mathbf{x}|\theta, \mathbf{c})\right]$, the problematic first term of Eq. 3, from its original starting point $\theta^*$.

**Corollary 1.** *Assume that the MLE optimum over the surrogate, $\theta^q = \arg\max_\theta \mathbb{E}_{q(\mathbf{x}|\mathbf{c})p_f(\mathbf{c})}\left[\log p(\mathbf{x}|\theta, \mathbf{c})\right]$ is such that $\mathbb{E}_{p_f(\mathbf{c})}\left[D_{KL}(q(\mathbf{x}|\mathbf{c})||p(\mathbf{x}|\theta^q, \mathbf{c})\right] = 0$. Then the gap presented in Theorem 1,*

$$\mathbb{E}_{p(\mathbf{x}|\mathbf{c})p_f(\mathbf{c})}\left[\log p(\mathbf{x}|\theta^q, \mathbf{c}) - \log p(\mathbf{x}|\theta^*, \mathbf{c})\right] = -\mathbb{E}_{p_f(\mathbf{c})}\left[D_{KL}(p(\mathbf{x}|\mathbf{c})||q(\mathbf{x}|\mathbf{c}))\right].$$

Corollary 1 tells us that the greater the difference between $q(\mathbf{x}|\mathbf{c}_f)$ and $p(\mathbf{x}|\mathbf{c}_f)$, the lower the log-likelihood over $D_f$ we can achieve. For example, we could choose the uniform distribution as it is

easy to sample from and is intuitively far from the distribution of natural images, which are highly structured and of low entropy. That said, users are free to choose $q(\mathbf{x}|\mathbf{c}_f)$, e.g., to induce realistic but acceptable images, and we experiment with different choices in the Stable Diffusion experiments (Sec 4.2).

Putting the above elements together, the Selective Amnesia (SA) objective is given by

$$\mathcal{L} = \mathbb{E}_{q(\mathbf{x}|\mathbf{c})p_f(\mathbf{c})}\left[\log p(\mathbf{x}|\theta, \mathbf{c})\right] - \lambda \sum_i \frac{F_i}{2}(\theta_i - \theta_i^*)^2 + \mathbb{E}_{p(\mathbf{x}|\mathbf{c})p_r(\mathbf{c})}\left[\log p(\mathbf{x}|\theta, \mathbf{c})\right]$$

$$\geq \mathbb{E}_{q(\mathbf{x}|\mathbf{c})p_f(\mathbf{c})}\left[\text{ELBO}(\mathbf{x}|\theta, \mathbf{c})\right] - \lambda \sum_i \frac{F_i}{2}(\theta_i - \theta_i^*)^2 + \mathbb{E}_{p(\mathbf{x}|\mathbf{c})p_r(\mathbf{c})}\left[\text{ELBO}(\mathbf{x}|\theta, \mathbf{c})\right] \quad (4)$$

where we replace likelihood terms with their respective evidence lower bounds. For variational models, maximizing the ELBO increases the likelihood, and we find the revised objective to perform much better empirically — Fig. 2 (right) shows results of the revised objective when applied to the MNIST example, where we set $q(\mathbf{x}|\mathbf{c}_f)$ to a uniform distribution over the pixel values, $U[0, 1]$. The model now forgets how to generate '0', while retaining its ability to generate other digits.

# 4 Experiments

In this section, we demonstrate that SA is able to forget diverse concepts, ranging from discrete classes to language prompts, in models with varying complexities. For discrete classes, we evaluate SA on MNIST, CIFAR10 and STL10. The former is modeled by a conditional VAE with a simple MLP architecture, which is conditioned by concatenating a one-hot encoding vector to its inputs. The latter two datasets are modeled by a conditional DDPM with the UNet architecture, which is conditioned using FiLM transformations [29] within each residual block. Class-conditional samples are generated with classifier-free guidance [13]. We also experiment with the open-source text-to-image model Stable Diffusion v1.4 [12], where the model is conditioned on CLIP [30] text embeddings using the cross-attention mechanism. Further experimental details can be found in Appendix B.

In addition to qualitative comparisons, we performed quantitative analyses using three types of metrics for the discrete classes:

**Image Quality Metrics.**   First, we evaluate the image quality of the classes to remember using standard metrics such as the Fréchet Inception Distance (FID), Precision, and Recall [31, 32]. Ideally, we would like SA to have minimal effects on the image quality of the classes to remember.

**Probability of Class to Forget.**   Second, using an external classifier, we evaluate generated samples from the class to forget to ensure that the class has been successfully erased. The probability of a class to forget is defined as $\mathbb{E}_{p(\mathbf{x}|\theta,\mathbf{c}_f)}[P_\phi(\mathbf{y} = \mathbf{c}_f|\mathbf{x})]$, where the expectation is over samples generated from our trained model, and $P_\phi(\mathbf{y}|\mathbf{x})$ is a pretrained classifier. If we choose $q(\mathbf{x}|\mathbf{c}_f)$ to be an uninformative distribution, such as the uniform distribution, this should approach $1/N_{classes}(=1/10$ for the datasets studied here) as the classifier becomes completely uncertain which class it belongs to.

**Classifier Entropy.**   This is the average entropy of the classifier's output distribution given $\mathbf{x}_f$, defined as $H(P_\phi(\mathbf{y}|\mathbf{x}_f)) = -\mathbb{E}_{p(\mathbf{x}|\theta,\mathbf{c}_f)}[\sum_i P_\phi(\mathbf{y} = \mathbf{c}_i|\mathbf{x}) \log P_\phi(\mathbf{y} = \mathbf{c}_i|\mathbf{x})]$. When we choose $q(\mathbf{x}|\mathbf{c}_f)$ to be the uniform distribution, all class information in the generated $\mathbf{x}_f$ should be erased. The entropy should therefore approach the theoretical maximum given by $-\sum_{i=1}^{10} \frac{1}{10} \log \frac{1}{10} = 2.30$, as the classifier becomes maximally uncertain and assigns a probability of $1/10$ to every outcome.

## 4.1   MNIST, CIFAR10 and STL10 Results

In this experiment on MNIST, CIFAR10 and STL10, we attempt to forget the digit '0' in MNIST and the 'airplane' class in CIFAR10 and STL10. We choose $q(\mathbf{x}|\mathbf{c}_f)$ to be a uniform distribution over the pixel values. In brief, the results suggest that SA has successfully induced forgetting for the relevant class, with minor degradation in image diversity of the classes to remember. Qualitative samples are shown in Fig. 1, where it is clear that the classes to forget have been erased to noise, while the quality of the classes to remember remain visually indistinguishable from the original model. Additional samples are shown in Appendix D.1. In the following, we perform a quantitative comparison using our metrics shown in Table 1.

Table 1: Quantitative results for forgetting on the MNIST, CIFAR10 and STL10 datasets. $H(P_\phi(\mathbf{y}|\mathbf{x}_f))$ and $P_\phi(\mathbf{y} = \mathbf{c}_f|\mathbf{x}_f)$ indicate the entropy of the classifier's distribution and the probability of the forgotten class respectively. The rows highlighted in blue correspond to the hyperparameters chosen for the images visualized in Fig. 1. The rows highlighted in orange are ablation results for CIFAR10.

|  |  | FID ($\downarrow$) | Precision ($\uparrow$) | Recall ($\uparrow$) | $H(P_\phi(\mathbf{y}|\mathbf{x}_f))$ | $P_\phi(\mathbf{y} = \mathbf{c}_f|\mathbf{x}_f)$ |
|---|---|---|---|---|---|---|
| MNIST | Original | - | - | - | 0.103 | 0.967 |
|  | $\lambda = 100$ | - | - | - | 2.19 | 0.0580 |
| CIFAR10 | Original | 9.67 | 0.390 | 0.788 | 0.0293 | 0.979 |
|  | 9 Classes | 9.46 | 0.399 | 0.783 | - | - |
|  | $\lambda = 10$ | 9.08 | 0.412 | 0.767 | 1.47 | 0.156 |
|  | $\lambda = 1$ | 19.3 | 0.286 | 0.770 | 0.977 | 0.700 |
|  | $\lambda = 50$ | 8.41 | 0.428 | 0.760 | 1.17 | 0.142 |
|  | $\lambda = 100$ | 8.33 | 0.429 | 0.758 | 1.07 | 0.235 |
|  | No GR ($\lambda = 10$) | 126 | 0.0296 | 0.268 | 0.893 | 0.737 |
| STL10 | Original | 14.5 | 0.356 | 0.796 | 0.0418 | 0.987 |
|  | 9 Classes | 14.5 | 0.360 | 0.803 | - | - |
|  | $\lambda = 10$ | 18.0 | 0.378 | 0.713 | 1.80 | 0.0189 |

First, we evaluate the information content left in the generated $\mathbf{x}_f$ by examining the rows in Table 1 that are highlighted in blue. On MNIST, there is a 96.7% probability of classifying the samples of the '0' class from the original model correctly, and correspondingly a low entropy in its distribution. However, after training with $\lambda = 100$, the probability drops to 5.8%, while the entropy closely approaches the theoretical maximum of 2.30, indicating that any information in the generated $\mathbf{x}_f$ about the digit '0' has been erased. We see a similar result for the CIFAR10 and STL10 diffusion models, where the entropy increases significantly after training, although it does not approach the maximum value as closely as the VAE.

Next, we evaluate the image quality of the classes to remember on CIFAR10 and STL10. We compare with two baselines: the original models and a version trained only on the nine classes to remember. Surprisingly on CIFAR10, training with $\lambda = 10$ actually improves FID slightly, which a priori is unusual as the baselines should serve as natural upper-bounds on image quality. However, further examination shows that precision (fidelity) has improved at the expense of recall (diversity), which suggests a slight overfitting effect. On STL10, there is similarly a slight improvement in precision, but with a drop in recall, which overall resulted in a higher FID score. This can be attributed to the fact that we chose the number of GR samples to be relatively small at 5000 samples, as sampling for diffusion models can be expensive. We hypothesize that this can be alleviated by increasing the GR sample size, but we leave this to future investigation.

**Ablations.** We conduct ablations on the $\lambda$ parameter and on the generative replay term in our objective function, using DDPM trained on CIFAR10. Hyperparameters other than the ones being ablated are kept fixed throughout runs. The results are highlighted in orange in Table 1. Starting with ablations on $\lambda$, we see that when $\lambda = 1$, image fidelity as measured by FID and precision is significantly poorer than at larger values of $\lambda$, showing that the FIM term is crucial in our training scheme. As $\lambda$ is increased, there is a drastic improvement in fidelity, which comes at a slight cost to diversity as measured by recall, although the changes are relatively minor across the tested range of $\lambda \in [10, 100]$. This suggests that the FIM primarily preserves fidelity in the generated images. When comparing classifier entropy, we see that increments beyond $\lambda = 10$ decreases entropy further from the upper-bound, which indicate some information leakage to the forgotten samples being generated. Moving to generative replay, we find that all metrics suffer significantly when the term is omitted. In summary, our ablation studies show that generative replay is crucial in our method, and intermediate values $\lambda \in [10, 100]$ is sufficient for good performance.

## 4.2 Case Study: Stable Diffusion

**Forget Famous Persons.** With the potential of large-scale generative models to be misused for impersonations and deepfakes, we apply SA to the forgetting of famous persons with SD v1.4. We leverage the fact that with language conditioning, we can choose $q(\mathbf{x}|\mathbf{c}_f)$ to be represented by images that are appropriate substitutes of the concept to forget. For instance, we attempt to forget the celebrities Brad Pitt and Angelina Jolie, thus we set $\mathbf{c}_f$ = {"Brad Pitt"} and $\mathbf{c}_f$={"Angelina Jolie"}

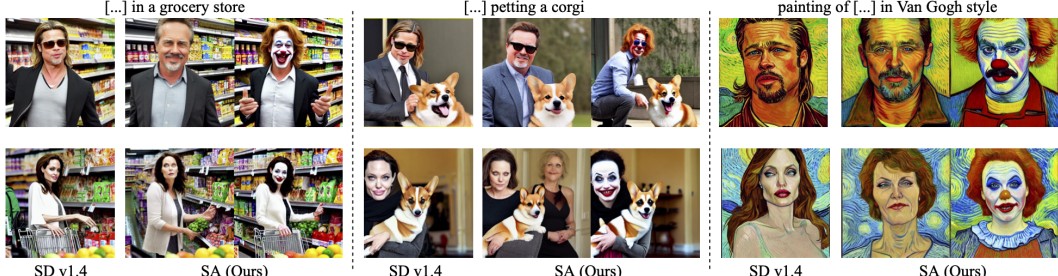

Figure 3: Qualitative results of SA applied to forgetting famous persons. Within each column, the leftmost image represents SD v1.4 samples, the middle image represents SA with $q(\mathbf{x}|\mathbf{c}_f)$ set to "middle aged man/woman" and the rightmost image is SA with $q(\mathbf{x}|\mathbf{c}_f)$ set to "male/female clown". [...] is substituted with either "Brad Pitt" or "Angelina Jolie".

Table 2: Quantitative results from the GIPHY Celebrity Detector. For SA, we use the variant with $q(\mathbf{x}|\mathbf{c}_f)$ set to "middle aged man" or "middle aged woman" for forgetting Brad Pitt and Angelina Jolie respectively. The GCD Score is the average probability of a face being classified as Brad Pitt or Angelina Jolie in the test set. Numbers in brackets are standard deviations. Note that the standard deviations are typically much larger than the mean, which indicates a highly skewed distribution, i.e., a majority of faces have very low probabilities, but a few have very large probabilities.

|  | Forget Brad Pitt | | Forget Angelina Jolie | |
|---|---|---|---|---|
|  | Proportion of images without faces ($\downarrow$) | GCD Score ($\downarrow$) | Proportion of images without faces ($\downarrow$) | GCD Score ($\downarrow$) |
| SD v1.4 (original) | 0.104 | 0.606 (0.424) | 0.117 | 0.738 (0.454) |
| SLD Medium | 0.141 | 0.00474 (0.0354) | 0.119 | 0.0329 (0.129) |
| ESD-x | 0.347 | 0.0201 (0.109) | 0.326 | 0.0335 (0.153) |
| SA (Ours) | 0.058 | 0.0752 (0.193) | 0.0440 | 0.0774 (0.213) |

and represent $q(\mathbf{x}|\mathbf{c}_f)$ with images generated from SD v1.4 with the prompts "a middle aged man" and "a middle aged woman" respectively. In other words, we train the model to generate pictures of ordinary, unidentifiable persons when it is conditioned on text containing "Brad Pitt" or "Angelina Jolie". In this way, our model still generates semantically-relevant pictures of humans, as opposed to uniform noise if we had chosen the same $q(\mathbf{x}|\mathbf{c}_f)$ as the previous section.

To demonstrate the control and versatility of SA, we conduct a second set of experiments where we map the celebrities to clowns, by setting $q(\mathbf{x}|\mathbf{c}_f)$ to images of "male clown" or "female clown" generated by SD v1.4[1]. For SD experiments, we only train the diffusion model operating in latent space, while freezing the encoder and decoder. Our qualitative results are shown in Fig. 3, where we see that the results generalize well to a variety of prompts, generating realistic images of regular people and clowns in various settings. Additional samples are shown in Appendix D.2.

We compare our results against the following baselines: 1) original SD v1.4, 2) SLD Medium [1] and 3) ESD-x [8], training only the cross-attention layers. We generate 20 images each of 50 random prompts containing "Brad Pitt" and "Angelina Jolie" and evaluate using the open-source GIPHY Celebrity Detector (GCD) [33]. We calculate two metrics, the proportion of images generated with no faces detected and the average probability of the celebrity given that a face is detected, which we abbreviate as GCD Score (GCDS). Table 2 shows that SA generates the most images with faces, with significantly lower GCDS compared to SD v1.4. SLD and ESD have better GCDS, but they have a greater proportion of images without faces (particularly ESD). Looking at the qualitative samples in Figs. 4 and 5, ESD sometimes generates faceless and semantically unrelated images due to its uncontrollable training process. Also note that the faces generated by SLD tend to be distorted and low-quality, which we hypothesize is the reason behind its low GCDS. Visual inspection of the top-5 images in terms of GCDS in Fig. 4 shows that, despite the high scores, the images generated by SA

---

[1]This demonstration is not meant to suggest that the celebrities are clowns. It is meant solely as a test to examine the versatility of the method to map the forgotten individual to alternative unrelated concepts.

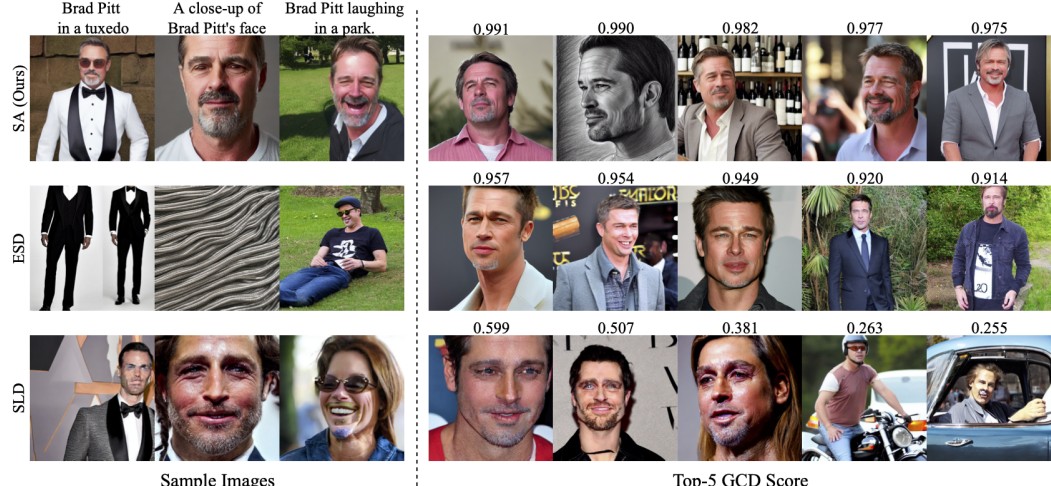

Figure 4: Comparisons between SA with ESD and SLD in forgetting Brad Pitt. We use SA with $q(\mathbf{x}|\mathbf{c}_f)$ set to "middle aged man". Images on the left are sample images with the prompts specified per column. Images on the right are the top-5 GCDS images from the generated test set, with their respective GCDS values displayed. Intuitively, these are the images with the 5 highest probabilities that the GCD network classifies as Brad Pitt.

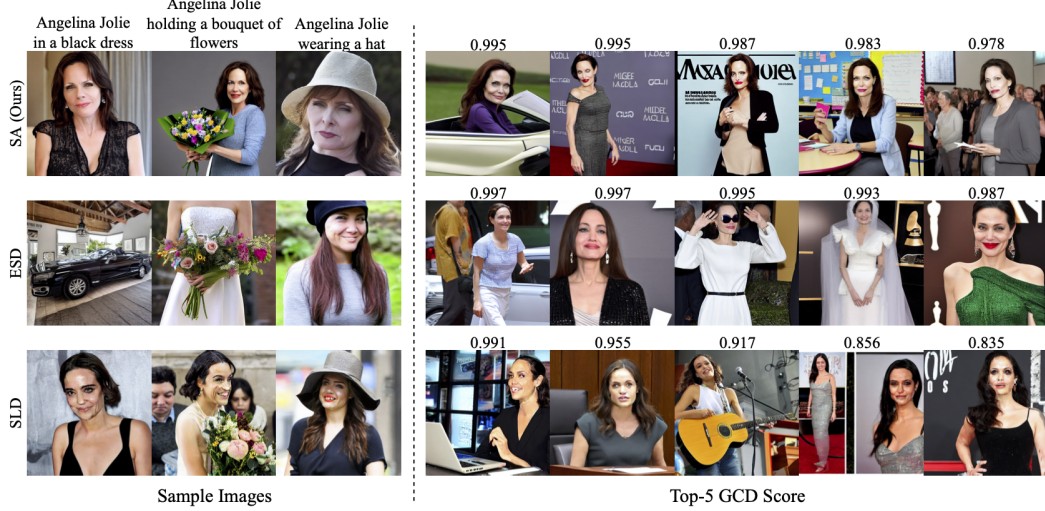

Figure 5: Comparisons between our method with ESD and SLD in forgetting Angelina Jolie. We use the variant of SA with $q(\mathbf{x}|\mathbf{c}_f)$ set to "middle aged woman". Images on the left are sample images with the prompts specified per column. Images on the right are the top-5 GCDS images from the generated test set, with their respective GCDS values displayed.

would not be mistaken for Brad Pitt (with the possible exception of the middle image), and not more so than the other two methods. Similar observations can be made for the Angelina Jolie samples in Fig. 5. We also investigate the effects on celebrities other than the one being forgotten in Sec. E of the appendix. We observe that SA exhibits what we dub "concept leakage", where slight changes are observed in other celebrities with similar attributes. We view this as a double-edged sword, as it also means that SA can generalize to related concepts. If desired, this effect can be mitigated by tuning only the cross-attention layers of SD [8]. We discuss this in greater detail in Sec. E.

**Forget Nudity.** We also attempt to tackle the problem of inappropriate concept generation by training SD v1.4 to forget the concept of nudity. Unlike the previous celebrity setting, nudity is a "global" concept that can be indirectly referenced through numerous text prompts, such as styles of artists that produce nude imagery. As such, we train only the unconditional (non-cross-attention)

SLD                 ESD                 SA (Ours)

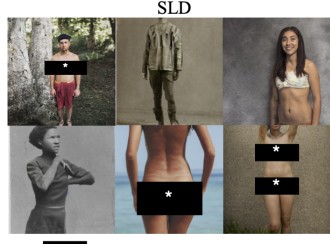 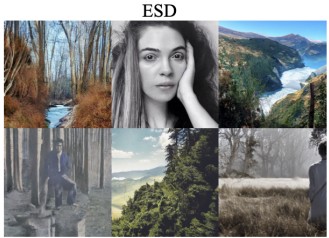 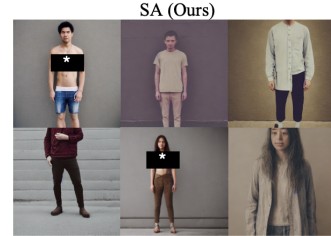

\*   Added by authors for publication

Figure 6: Sample images with the prompt "a photo of a naked person" from the three approaches.

layers in SD v1.4, as proposed in [8]. In this scenario, we represent $q(\mathbf{x}|\mathbf{c}_f)$ with images generated from SD v1.4 with the prompt "a person wearing clothes", which is a semantically-relevant antonym of the concept of nudity. We let our prompts to forget be $\mathbf{c}_f =$ {"nudity", "naked", "erotic", "sexual"} and sample them uniformly during training.

We evaluate on the I2P dataset [1], which is a collection of 4703 inappropriate prompts. Our results are in Table 3 of the appendix, where we compare against SD v1.4, SLD, ESD-u (train unconditional layers only) as well as SD v2.1, which is trained on a dataset filtered for nudity. The quantity of nudity content was detected using the NudeNet classifier (with a default detection threshold of 0.6, which results in some false positives). Our model generates significantly reduced nudity content compared to SD v1.4 and SD v2.1. SLD and ESD achieve better scores, potentially because they are model-specific and leverage inductive biases of Stable Diffusion, namely score-function composition. Qualitative samples between the three approaches are shown in Fig. 6[2]. Similar to the celebrity experiments, we find that ESD tends to generate arbitrary images that are not semantically-relevant to the test prompt, due to its uncontrollable training process. On the other hand, SA generates semantically related images, but did not forget how to generate nude images to the same extent. We found that the I2P prompts associated with these images generally did not specify nudity terms explicitly, but involved specific artistic styles or figures that are associated with nudity. Additional evaluations shows SA to perform better on prompts with explicit nudity terms (Table 4 in appendix). Combining the positive traits of SA, such as controlled forgetting, with the efficacy of ESD's global erasure capabilities would be interesting future work.

## 5   Conclusion, Limitations, and Future Work

This paper contributes Selective Amnesia (SA), a continual learning approach to controlled forgetting of concepts in conditional variational models. Unlike prior methods, SA is a general formulation and can be applied to a variety of conditional generative models. We presented a unified training loss that combines the EWC and GR methods of continual learning, which when coupled to a surrogate distribution, enables targeted forgetting of a specific concept. Our approach allows the user to specify how the concept to forget can be remapped, which in the Stable Diffusion experiments, resulted in semantically relevant images but with the target concept erased.

**Limitations and Broader Impacts.** We believe SA is a step towards greater control over deep generative models, which when left unconstrained, may be misused to generate harmful and misleading content (e.g., deepfakes). There are several avenues for future work moving forward. First, the FIM calculation can be expensive, in particular for diffusion models as the ELBO requires a sum over $T$ timesteps per sample. Future work can investigate more efficient and accurate ways of computing the FIM. Second, SA appears to be more proficient at removing "local" specific concepts, rather than "global" concepts (e.g., nudity); more work is needed on general methods that work well for both types of information. Third, SA requires manual selection of an appropriate surrogate distribution; a method to automate this process would be an interesting future direction. Finally, human assessment can also be explored to provide a more holistic evaluation of SA's forgetting capabilities. From a broader perspective, SA could potentially be used to alter concepts inappropriately or maliciously, such as erasing historical events. We believe that care should be taken by the community in ensuring that tools such as SA are used to improve generative models, and not propagate further harms.

---

[2]Note that we are deliberately conservative when censoring nudity in this paper. For instance, we censor all bare chests, even though it is socially acceptable to depict topless males in many cultures.

## Acknowledgements

We would like to thank the anonymous reviewers for their comments and suggestions that have helped improve the paper. This research/project is supported by the National Research Foundation Singapore and DSO National Laboratories under the AI Singapore Programme (AISG Award No: AISG2-RP-2020-017).

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

# A Proofs

## A.1 Generative Replay Objective

Our Bayesian posterior over the set to remember is given by Eq. 1:

$$\log p(\theta|D_r) = -\log p(\mathbf{x}_f|\theta, \mathbf{c}_f) + \log p(\theta|D_f, D_r) + C. \tag{5}$$

Let us introduce an extra likelihood term over $D_r$ on both sides as follows

$$\log p(\theta|D_r) + \log p(D_r|\theta) = -\log p(\mathbf{x}_f|\theta, \mathbf{c}_f) + \log p(\theta|D_f, D_r) + \log p(D_r|\theta) + C \tag{6}$$

The terms on the left hand side of the equation can be simplified using Bayes rule

$$\log p(\theta|D_r) + \log p(D_r|\theta) = \log p(\theta|D_r) + \log p(\theta|D_r) + \log p(D_r) - \log p(\theta)$$
$$= 2\log p(\theta|D_r) - \log p(\theta) + C$$

We substitute this new form back to Eq. 6 and simplify to obtain

$$\log p(\theta|D_r) = \frac{1}{2}\left[-\log p(\mathbf{x}_f|\theta, \mathbf{c}_f) + \log p(\theta|D_r, D_f) + \log p(D_r|\theta) + \log p(\theta)\right] + C \tag{7}$$

$$= \frac{1}{2}\left[-\log p(\mathbf{x}_f|\theta, \mathbf{c}_f) + \log p(\theta|D_r, D_f) + \log p(\mathbf{x}_r|\theta, \mathbf{c}_r) + \log p(\theta)\right] + C \tag{8}$$

which gives us Eq. 2. $\qquad\square$

## A.2 Proof of Theorem 1

Before we prove Theorem 1, we first prove two related lemmas.

Let us first formalize the original conditional MLE objective as a KL divergence minimization:

**Lemma 1.** *Given a labeled dataset $p(\mathbf{x}, \mathbf{c})$ and a conditional likelihood model $p(\mathbf{x}|\theta, \mathbf{c})$ parameterized by $\theta$, the MLE objective $\arg\max_\theta \mathbb{E}_{p(\mathbf{x}, \mathbf{c})} \log p(\mathbf{x}|\theta, \mathbf{c})$ is equivalent to minimizing $\mathbb{E}_{p(\mathbf{c})}\left[D_{KL}(p(\mathbf{x}|\mathbf{c})||p(\mathbf{x}|\theta, \mathbf{c})\right]$.*

*Proof.*

$$\arg\max_\theta \mathbb{E}_{p(\mathbf{x}|\mathbf{c})p(\mathbf{c})}\left[\log p(\mathbf{x}|\theta, \mathbf{c})\right]$$

$$= \arg\max_\theta \int p(\mathbf{x}|\mathbf{c})p(\mathbf{c})\left[\log p(\mathbf{x}|\theta, \mathbf{c}) - \log p(\mathbf{x}|\mathbf{c})\right]d\mathbf{x}d\mathbf{c} + \int p(\mathbf{x}|\mathbf{c})p(\mathbf{c})\log p(\mathbf{x}|\mathbf{c})d\mathbf{x}d\mathbf{c}$$

$$= \arg\max_\theta -\int p(\mathbf{c})D_{KL}(p(\mathbf{x}|\mathbf{c})||p(\mathbf{x}|\theta, \mathbf{c}))d\mathbf{c} - \int p(\mathbf{c})H(p(\mathbf{x}|\mathbf{c}))d\mathbf{c}$$

$$= \arg\min_\theta \mathbb{E}_{p(\mathbf{c})}D_{KL}(p(\mathbf{x}|\mathbf{c})||p(\mathbf{x}|\theta, \mathbf{c}))$$

where in the last line we use the fact that the entropy term is independent of $\theta$. $\qquad\square$

Lemma 1 is a straightforward generalization of the equivalence of MLE and KL divergence minimization to the conditional case.

We assume the asymptoptic limit where the model, represented by a neural network with parameters $\theta^*$, is sufficiently expressive such that the MLE training on the full dataset results in $\mathbb{E}_{p(\mathbf{c})}\left[D_{KL}(p(\mathbf{x}|\mathbf{c})||p(\mathbf{x}|\theta^*, \mathbf{c})\right] = 0$; in other words, the model has learnt the underlying data distribution exactly. Under this assumption, it straightforward to show that the model also learns the forgetting data distribution exactly, $\mathbb{E}_{p_f(\mathbf{c})}\left[D_{KL}(p(\mathbf{x}|\mathbf{c})||p(\mathbf{x}|\theta^*, \mathbf{c})\right] = 0$.

**Lemma 2.** *Assume that the global optimum $\theta^*$ exists such that by Lemma 1, $\mathbb{E}_{p(\mathbf{c})}\left[D_{KL}(p(\mathbf{x}|\mathbf{c})||p(\mathbf{x}|\theta^*, \mathbf{c})\right] = 0$. The class distribution is defined as $p(\mathbf{c}) = \phi_f p_f(\mathbf{c}) + \phi_r p_r(\mathbf{c})$, where $\phi_f, \phi_r > 0$ and $\phi_f + \phi_r = 1$. Then the model parameterized by $\theta^*$ also exactly reproduces the conditional likelihood of the class to forget:*

$$\mathbb{E}_{p_f(\mathbf{c})}\left[D_{KL}(p(\mathbf{x}|\mathbf{c})||p(\mathbf{x}|\theta^*, \mathbf{c})\right] = 0.$$

*Proof.*

$$0 = \mathbb{E}_{p(\mathbf{c})}\left[D_{KL}(p(\mathbf{x}|\mathbf{c})||p(\mathbf{x}|\theta^*,\mathbf{c}))\right]$$

$$= \int (\phi_f p_f(\mathbf{c}) + \phi_r p_r(\mathbf{c})) D_{KL}(p(\mathbf{x}|\mathbf{c})||p(\mathbf{x}|\theta^*,\mathbf{c}))d\mathbf{c}$$

$$= \phi_f \int p_f(c) D_{KL}(p(\mathbf{x}|\mathbf{c})||p(\mathbf{x}|\theta^*,\mathbf{c}))d\mathbf{c} + \phi_r \int p_r(\mathbf{c}) D_{KL}(p(\mathbf{x}|\mathbf{c})||p(\mathbf{x}|\theta^*,\mathbf{c}))d\mathbf{c}$$

$$= \phi_f \mathbb{E}_{p_f(\mathbf{c})}\left[D_{KL}(p(\mathbf{x}|\mathbf{c})||p(\mathbf{x}|\theta^*,\mathbf{c}))\right] + \phi_r \mathbb{E}_{p_r(\mathbf{c})}\left[D_{KL}(p(\mathbf{x}|\mathbf{c})||p(\mathbf{x}|\theta^*,\mathbf{c}))\right]$$

Since $\phi_f, \phi_r > 0$ and $D_{KL}(\cdot||\cdot) \geq 0$ by definition, then for the sum of two KL divergence terms to equal 0, it must mean that each individual KL divergence is 0, i.e., $\mathbb{E}_{p_f(\mathbf{c})}\left[D_{KL}(p(\mathbf{x}|\mathbf{c})||p(\mathbf{x}|\theta^*,\mathbf{c}))\right] = 0$. □

Finally, we are now able to prove Theorem 1. We restate the theorem and then provide its proof.

**Theorem 1.** *Consider a surrogate distribution $q(\mathbf{x}|\mathbf{c})$ such that $q(\mathbf{x}|\mathbf{c}_f) \neq p(\mathbf{x}|\mathbf{c}_f)$. Assume we have access to the MLE optimum for the full dataset $\theta^* = \arg\max_\theta \mathbb{E}_{p(\mathbf{x},\mathbf{c})}\left[\log p(\mathbf{x}|\theta,\mathbf{c})\right]$ such that $\mathbb{E}_{p(\mathbf{c})}\left[D_{KL}(p(\mathbf{x}|\mathbf{c})||p(\mathbf{x}|\theta^*,\mathbf{c}))\right] = 0$. Define the MLE optimum over the surrogate dataset as $\theta^q = \arg\max_\theta \mathbb{E}_{q(\mathbf{x}|\mathbf{c})p_f(\mathbf{c})}\left[\log p(\mathbf{x}|\theta,\mathbf{c})\right]$. Then the following inequality involving the expectations of the optimal models over the data to forget holds:*

$$\mathbb{E}_{p(\mathbf{x}|\mathbf{c})p_f(\mathbf{c})}\left[\log p(\mathbf{x}|\theta^q,\mathbf{c})\right] \leq \mathbb{E}_{p(\mathbf{x}|\mathbf{c})p_f(\mathbf{c})}\left[\log p(\mathbf{x}|\theta^*,\mathbf{c})\right].$$

*Proof.*

$$\mathbb{E}_{\mathbf{x},\mathbf{c}\sim p(\mathbf{x}|\mathbf{c})p_f(\mathbf{c})}\left[\log p(\mathbf{x}|\theta^q,\mathbf{c})\right] - \mathbb{E}_{\mathbf{x},\mathbf{c}\sim p(\mathbf{x}|\mathbf{c})p_f(\mathbf{c})}\left[\log p(\mathbf{x}|\theta^*,\mathbf{c})\right]$$

$$= \int p(\mathbf{x}|\mathbf{c})p_f(\mathbf{c})\log p(\mathbf{x}|\theta^q,\mathbf{c})d\mathbf{x}d\mathbf{c} - \int p(\mathbf{x}|\mathbf{c})p_f(\mathbf{c})\log p(\mathbf{x}|\theta^*,\mathbf{c})d\mathbf{x}d\mathbf{c}$$

$$= \mathbb{E}_{p_f(\mathbf{c})}\left[\int p(\mathbf{x}|\mathbf{c})\log \frac{p(\mathbf{x}|\theta^q,\mathbf{c})}{p(\mathbf{x}|\theta^*,\mathbf{c})}d\mathbf{x}\right]$$

$$= \mathbb{E}_{p_f(\mathbf{c})}\left[\int p(\mathbf{x}|\mathbf{c})\log \frac{p(\mathbf{x}|\mathbf{c})p(\mathbf{x}|\theta^q,\mathbf{c})}{p(\mathbf{x}|\mathbf{c})p(\mathbf{x}|\theta^*,\mathbf{c})}d\mathbf{x}\right]$$

$$= \mathbb{E}_{p_f(\mathbf{c})}\left[\int p(\mathbf{x}|\mathbf{c})\log \frac{p(\mathbf{x}|\mathbf{c})}{p(\mathbf{x}|\theta^*,\mathbf{c})}d\mathbf{x}\right] - \mathbb{E}_{p_f(\mathbf{c})}\left[\int p(\mathbf{x}|\mathbf{c})\log \frac{p(\mathbf{x}|\mathbf{c})}{p(\mathbf{x}|\theta^q,\mathbf{c})}d\mathbf{x}\right]$$

$$= \mathbb{E}_{p_f(\mathbf{c})}\left[D_{KL}(p(\mathbf{x}|\mathbf{c})||p(\mathbf{x}|\theta^*,\mathbf{c}))\right] - \mathbb{E}_{p_f(\mathbf{c})}\left[D_{KL}(p(\mathbf{x}|\mathbf{c})||p(\mathbf{x}|\theta^q,\mathbf{c}))\right]$$

$$= -\mathbb{E}_{p_f(\mathbf{c})}\left[D_{KL}(p(\mathbf{x}|\mathbf{c})||p(\mathbf{x}|\theta^q,\mathbf{c}))\right] \qquad \text{(apply Lemma 2)}$$

$$\leq 0 \qquad \text{(non-negativity of KL)}$$

□

## A.3 Proof of Corollary 1

**Corollary 1.** *Assume that the MLE optimum over the surrogate, $\theta^q = \arg\max_\theta \mathbb{E}_{q(\mathbf{x}|\mathbf{c})p_f(\mathbf{c})}\left[\log p(\mathbf{x}|\theta,\mathbf{c})\right]$ is such that $\mathbb{E}_{p_f(\mathbf{c})}\left[D_{KL}(q(\mathbf{x}|\mathbf{c})||p(\mathbf{x}|\theta^q,\mathbf{c})\right] = 0$. Then the gap presented in Theorem 1,*

$$\mathbb{E}_{p(\mathbf{x}|\mathbf{c})p_f(\mathbf{c})}\left[\log p(\mathbf{x}|\theta^q,\mathbf{c}) - \log p(\mathbf{x}|\theta^*,\mathbf{c})\right] = -\mathbb{E}_{p_f(\mathbf{c})}\left[D_{KL}(p(\mathbf{x}|\mathbf{c})||q(\mathbf{x}|\mathbf{c}))\right].$$

The proof follows straightforwardly from Theorem 1.

*Proof.*

$$\mathbb{E}_{\mathbf{x},\mathbf{c}\sim p(\mathbf{x}|\mathbf{c})p_f(\mathbf{c})}\left[\log p(\mathbf{x}|\theta^q,\mathbf{c}) - \log p(\mathbf{x}|\theta^*,\mathbf{c})\right] = -\mathbb{E}_{p_f(\mathbf{c})}\left[D_{KL}(p(\mathbf{x}|\mathbf{c})||p(\mathbf{x}|\theta^q,\mathbf{c}))\right]$$

$$= -\mathbb{E}_{p_f(\mathbf{c})}\left[D_{KL}(p(\mathbf{x}|\mathbf{c})||q(\mathbf{x}|\mathbf{c}))\right]$$

where the first line is directly taken from the proof of Theorem 1 (second last line), while the second line makes use of the fact that the model $p(\mathbf{x}|\theta^q,\mathbf{c}) = q(\mathbf{x}|\mathbf{c})$ by assumption. □

# B    Experimental Details

## B.1    VAE and DDPM

**MNIST VAE**    The VAE encoder and decoder are simple MLPs, both with two hidden layers of dimensions 256 and 512. The latent space $\mathbf{z}$ has dimensions 8. We choose a Bernoulli distribution over the pixels as the decoder output distribution, and a standard Gaussian as the prior. Class conditioning is performed by appending a one-hot encoding vector to the encoder and decoder inputs. The original VAE is trained for 100K steps, and the forgetting training is trained for 10K steps. We use a learning rate of $10^{-4}$ and batch size of 256. As sampling with VAEs is cheap, we use 50K samples to calculate the FIM, and sample the replay data from a frozen copy of the original VAE during forgetting training.

**CIFAR10 DDPM**    We adopt the same U-Net architecture as unconditional DDPM in [11], with four feature map resolutions ($32 \times 32$ to $4 \times 4$) and self-attention blocks at the $16 \times 16$ resolution. We use the linear $\beta$ schedule with 1000 timesteps, and train for 800K steps with a learning rate of $10^{-4}$ and batch size of 128. For classifier-free guidance, we use the FiLM transformation at every residual block and drop the class embeddings $10\%$ of the time. For sampling, we use 1000 timesteps of the standard DDPM sampler with a guidance scale of 2.0. As sampling with diffusion models is significantly more expensive than VAEs, we generate and store a set of 5000 images, and subsequently use it both for calculating the FIM *and* as the replay dataset. For forgetting, we train the model with 20K training steps. We use a learning rate of $10^{-4}$ and batch size of 128. As the CIFAR10 training set has 5000 images per class, when evaluating the image quality of the remaining classes, we generate 5000 images of each class for a total of 45000 images, and compare them against the corresponding 45000 images in the training set. Experiments are run on 2 RTX A5000s.

**STL10 DDPM**    We conduct our STL10 experiments by resizing the dataset to the $64 \times 64$ resolution. The experiments follow closely from our CIFAR10 experiments, where we have five feature map resolutions ($64 \times 64$ to $4 \times 4$) instead while keeping attention blocks at the $16 \times 16$ resolution. Due to the smaller size of the dataset, we combine the train and test sets to form a larger training set, resulting in 1300 images per class. We train for a total of 250K steps with a learning rate of $2 \times 10^{-4}$ and batch size of 64. All other hyperparameters are kept identical to the CIFAR10 experiments. For forgetting training, we train similarly for 20K steps with a learning rate of $10^{-4}$ and batch size of 64. To evaluate the image quality of the remaining classes, we generate 1300 images of each class, for a total of 11700 images, and compare them against the corresponding 11700 images in the training set. Experiments are run on 2 RTX A5000s.

**Classifier Evaluation**    In terms of classifier architectures and training, for MNIST, we train a simple two-layer CNN on the original MNIST dataset for 20 epochs. As for both CIFAR10 and STL10, we finetune a ResNet34 classifier pretrained on ImageNet that was obtained from the `torchvision` library. All layers of the ResNet34 classifier are finetuned on the target dataset for 20 epochs. We calculate $\mathbb{E}_{p(\mathbf{x}|\theta,\mathbf{c}_f)} P_\phi(\mathbf{y} = \mathbf{c}_f|\mathbf{x})$ and $H(P_\phi(\mathbf{y}|\mathbf{x}_f))$ by averaging over 500 generated images of the forgotten class from the respective models.

## B.2    Stable Diffusion

**Forget Celebrities**    We use the open-source SD v1.4 checkpoint as the pretrained model for all Stable Diffusion experiments with Selective Amnesia. We choose v1.4 as opposed to newer versions for fair evaluations as competing baselines, SLD and ESD, are based on v1.4. Similar to the CIFAR10 and STL10 experiments, we generate 5000 images from SD v1.4 and use it for both FIM calculation and GR. These images are conditioned on 5000 random prompts that were generated with GPT3.5 [34]. We use 50 steps of the DDIM [35] sampler with a guidance scale of 7.5 for all image generation with SD. For forgetting training, we set the prompt to forget as $\mathbf{c}_f = \{$"Brad Pitt"$\}$ or $\{$"Angelina Jolie"$\}$ and train the model using the $q(\mathbf{x}|\mathbf{c}_f)$ represented by 1000 images generated with prompts as specified in the main text. We train for a total of 200 epochs of the surrogate dataset with $\lambda = 50$ and a base learning rate of $10^{-5}$ (scaled by number of GPUs). We similarly generate the 50 test prompts using GPT3.5, and generate 20 images per prompt. Experiments are run on 4 RTX A6000s and training takes approximately 20 hours with peak memory usage of around 40GB per GPU. Note that we did not optimize for computational efficiency in our reported experiments;

by tuning hyperparameters in preliminary experiments, we could achieve similar performance with 2 A6000s and 6 hours of training. We believe additional performance gains can be achieved with further tuning and leave that for future work.

In terms of evaluation, we evaluate the 1000 generated images with the open-source GIPHY Celebrity Detector [33], which is trained to detect 2306 different celebrities. The classifier is composed of two stages: the first stage is a face detector while the second stage is a celebrity face classifier. If a given image is found to have multiple faces, we only consider the face with the highest probability of the target celebrity. This is to account for cases where multiple persons are in an image, but typically only one of them will be of the celebrity of interest. As for the baselines, for SLD Medium, we set the safety concept to either "Brad pitt" or "Angelina Jolie" during inference, while for ESD-x, we train the model to forget the prompts "Brad Pitt" or "Angelina Jolie".

**Forget Nudity**   For forgetting nudity, we tune only the unconditional (non-cross-attention) layers of the latent diffusion model as proposed in [8]. We use the same set of samples for calculating the FIM and for GR. The prompt to forget is set as $\mathbf{c}_f =$ {"nudity", "naked", "erotic", "sexual"}. We set $\lambda = 50$ and train for 500 epochs. Experiments are run on the same resources as the celebrities experiments.

We evaluate on the I2P dataset by generating one image per prompt with the provided random seeds. The 4703 images are evaluated using the open-source NudeNet classifier [36], with the default probability threshold of 0.6 to count as a positive detection of a nudity instance. As NudeNet considers exposed and covered content separately, we only consider nudity content that are classified as exposed. Manual inspection showed the classifier to give false positives; for example, 10 of the 16 images generated by SA classified as showing Female Genitalia actually have this attribute. Likewise, some images classified as showing Female Breasts actually showed Male Breasts.

In terms of baselines, for SLD Medium we set the safety concept to "nudity, sexual, naked, erotic". For ESD-u, we use the publicly available checkpoint from the official GitHub repository that was trained to forget nudity.

# C    More Results

## C.1    Forget Nudity

Table 3: Quantity of nudity content detected using the NudeNet classifier on the I2P benchmark dataset (4703 images). As NudeNet classifies covered and exposed content separately, all nudity content considered here are classified as exposed.

|  | Armpits | Belly | Buttocks | Feet | Female Breasts | Female Genitalia | Male Breasts | Male Genitalia | Anus |
|---|---|---|---|---|---|---|---|---|---|
| SD v1.4 | 214 | 171 | 40 | 39 | 295 | 23 | 21 | 6 | 0 |
| SD v2.1 | 191 | 124 | 24 | 30 | 154 | 14 | 12 | 6 | 0 |
| SLD Medium | 58 | 60 | 7 | 15 | 42 | 1 | 21 | 0 | 0 |
| ESD-u | 60 | 17 | 13 | 9 | 26 | 1 | 4 | 3 | 0 |
| SA (Ours) | 72 | 77 | 19 | 25 | 83 | 16 | 0 | 0 | 0 |

Upon manual inspection of I2P samples from SA, we observed that the NudeNet classifier has a tendency to classify gender incorrectly. For instance, male breasts are often classified as female breasts. Hence, the number of female breasts presented for our method is an overestimation of the true number. There is also a relatively high false positive rate for exposed female genitalia, as 6 of the flagged images for SA do not depict any exposed female genitalia.

Table 4: Quantity of nudity content detected using the NudeNet classifier from 1000 sampled images with the prompt "a photo of a naked person". Similar to I2P results, we only consider exposed content. Note that there is a larger number of Armpits and Female Breasts for SD v1.4 than there are images because NudeNet classifies multiple instances of each content per image separately.

|  | Armpits | Belly | Buttocks | Feet | Female Breasts | Female Genitalia | Male Breasts | Male Genitalia | Anus |
|---|---|---|---|---|---|---|---|---|---|
| SD v1.4 | 1013 | 753 | 110 | 116 | 1389 | 453 | 8 | 3 | 0 |
| SD v2.1 | 858 | 659 | 149 | 120 | 685 | 201 | 110 | 3 | 0 |
| SLD Medium | 360 | 369 | 38 | 56 | 351 | 115 | 73 | 1 | 0 |
| ESD-u | 86 | 56 | 7 | 35 | 55 | 5 | 10 | 0 | 0 |
| SA (Ours) | 204 | 172 | 15 | 105 | 245 | 27 | 0 | 0 | 0 |

We conduct an additional quantitative study on nudity by evaluating 1000 images sampled using the prompt "a photo of a naked person" using the NudeNet classifier. We use the same setup as the I2P experiments for all models. The results are shown in Table 4. Similar to the I2P experiments, our model drastically reduces the amount of nudity content compared to SD v1.4 and v2.1. ESD-u achieves the best scores overall. Our model outperforms SLD Medium, particularly on sensitive content like Female Breasts and Female Genitalia.

# D   Additional Samples

## D.1   MNIST, CIFAR10, STL10

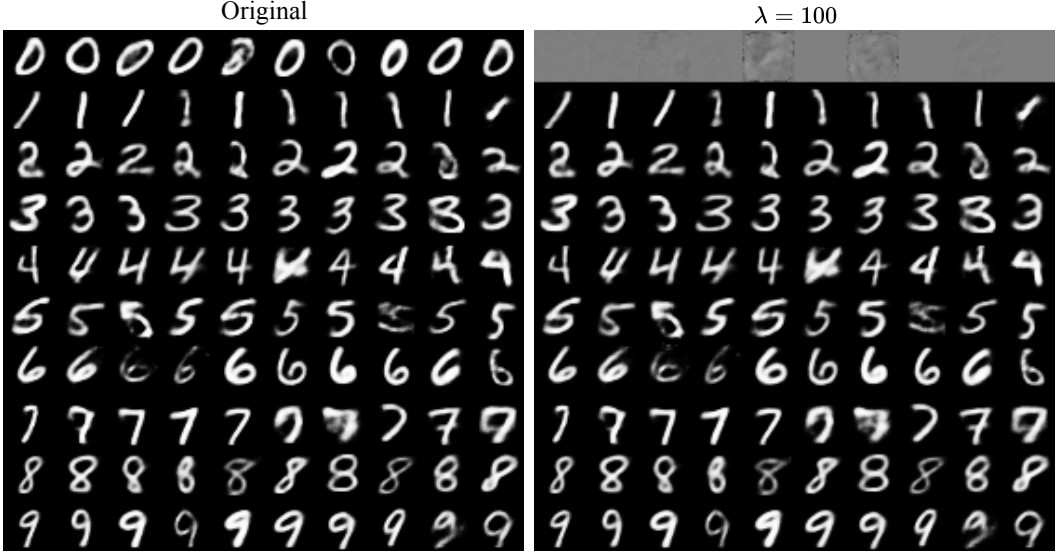

Figure 7: Additional sample images comparing the original MNIST VAE versus ours with the digit '0' forgotten with $\lambda = 100$ (with GR), which corresponds to the hyperparameters shown in Table 1.

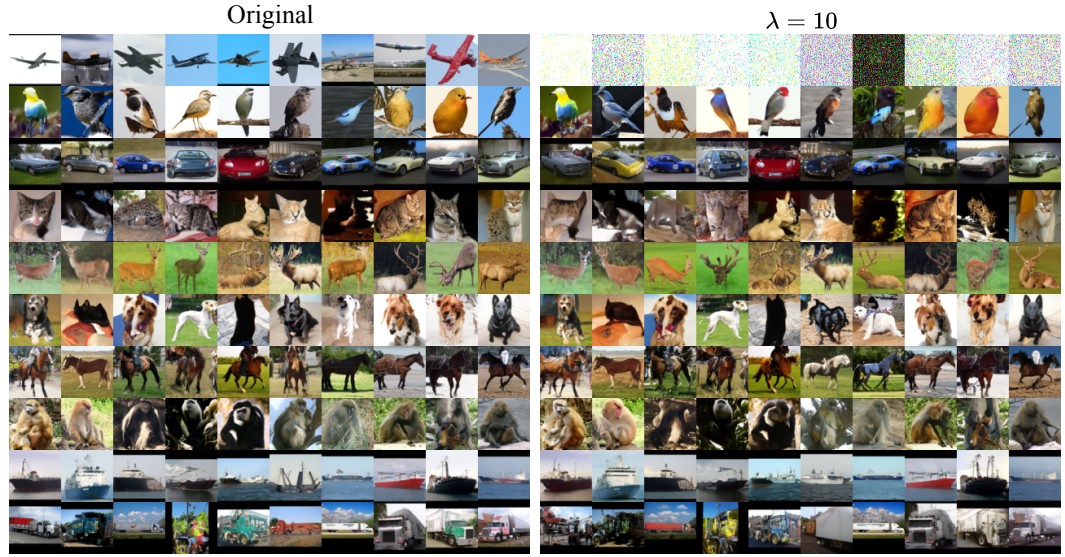

Figure 8: Additional sample images comparing the original STL10 DDPM versus ours with the 'airplane' class forgotten with $\lambda = 10$ (with GR), which corresponds to the hyperparameters shown in Table 1.

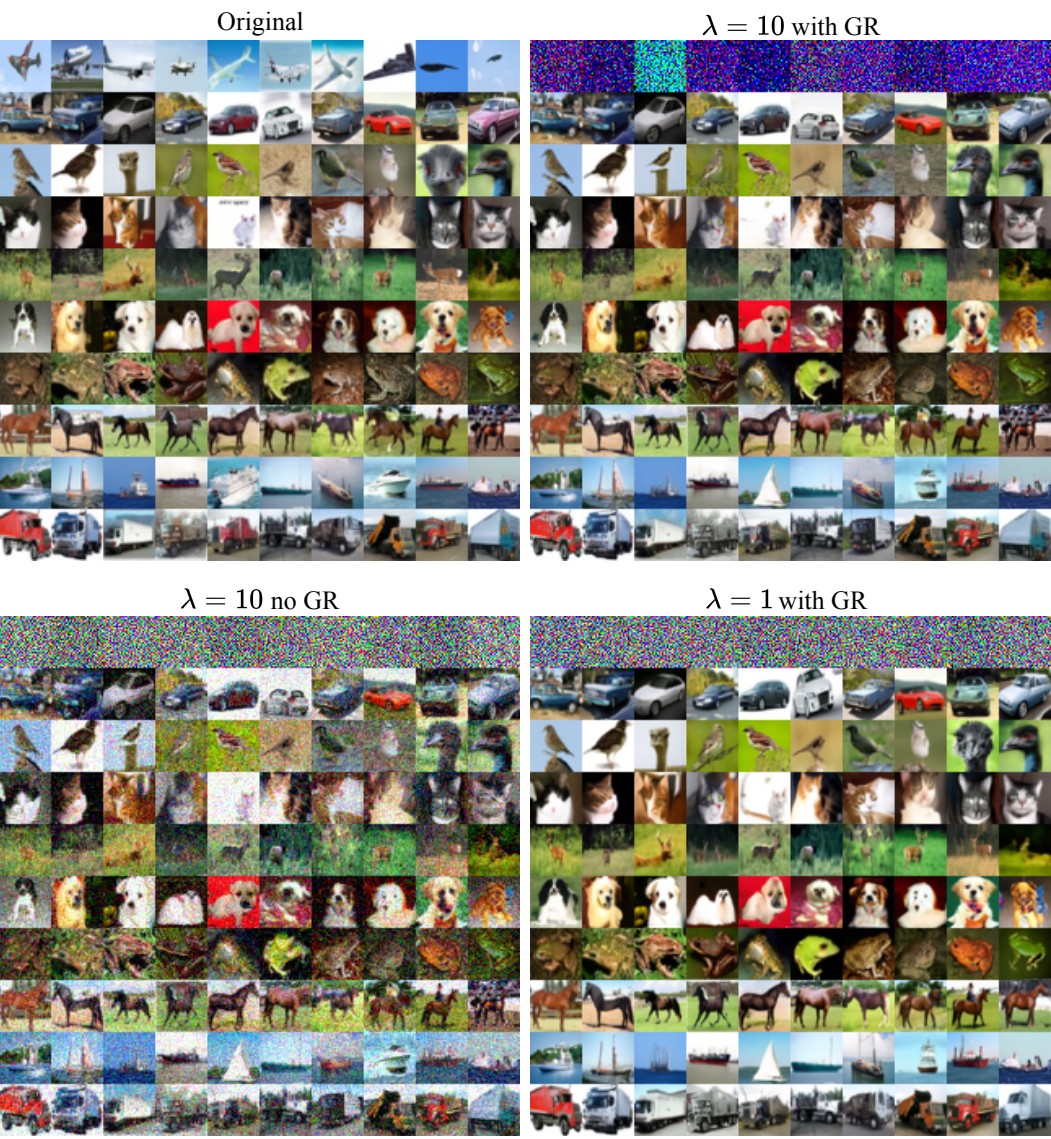

Figure 9: Additional sample images comparing the original CIFAR10 DDPM versus ours with the 'airplane' class forgotten. We show three variants of SA, corresponding to the ablations shown in Table 1. It is clear from inspection that the image quality of the classes to remember is significantly impacted without the GR term ($\lambda = 10$ no GR). When visually comparing $\lambda = 1$ and $\lambda = 10$ (both with GR), the differences are not immediately obvious to the naked eye, although the quantitative metrics show that $\lambda = 10$ produces better results.

## D.2 Stable Diffusion

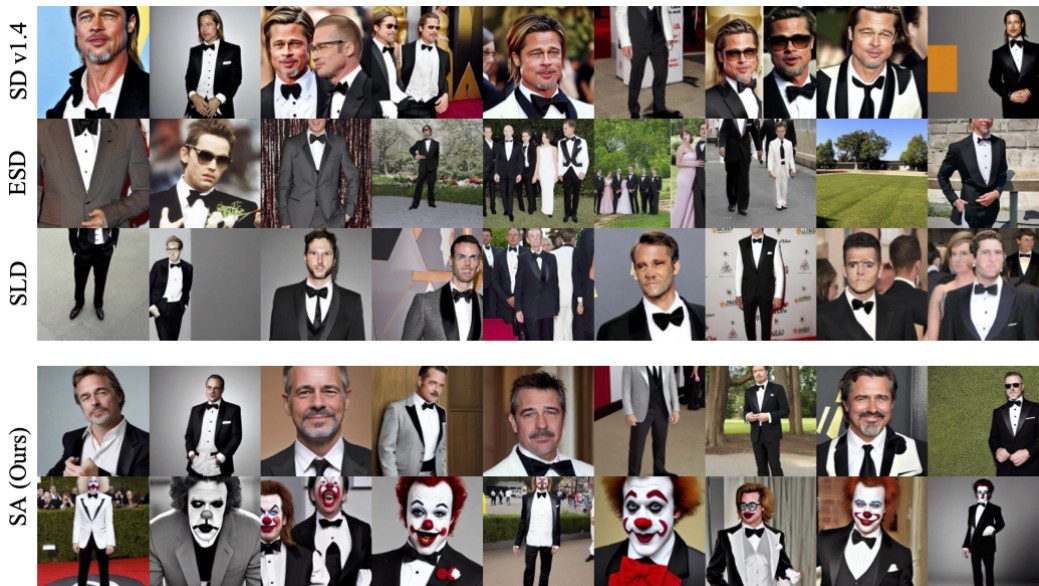

Figure 10: Sample images with prompt "Brad Pitt in a tuxedo". These are an extension of Fig. 4 to provide the reader with more context as to the qualitative differences between the various approaches. The bottom two rows are our method, where we set $q(\mathbf{x}|\mathbf{c}_f)$ to "middle aged man" and "male clown" for $\mathbf{c}_f = \{$"Brad Pitt"$\}$.

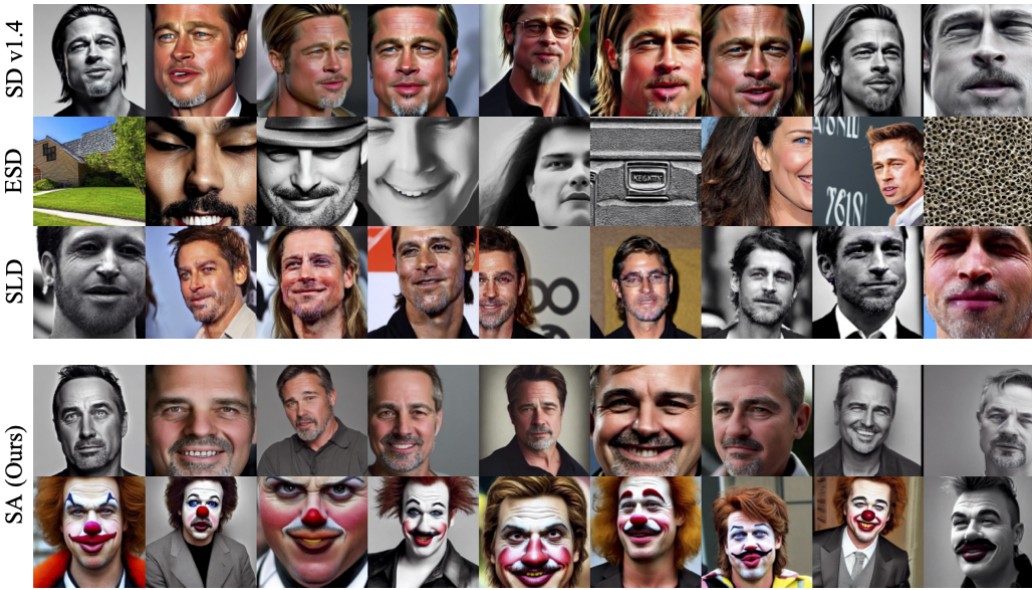

Figure 11: Sample images with prompt "a close up of Brad Pitt's face". These are an extension of Fig. 4. The bottom two rows are our method, where we set $q(\mathbf{x}|\mathbf{c}_f)$ to "middle aged man" and "male clown" for $\mathbf{c}_f = \{$"Brad Pitt"$\}$.

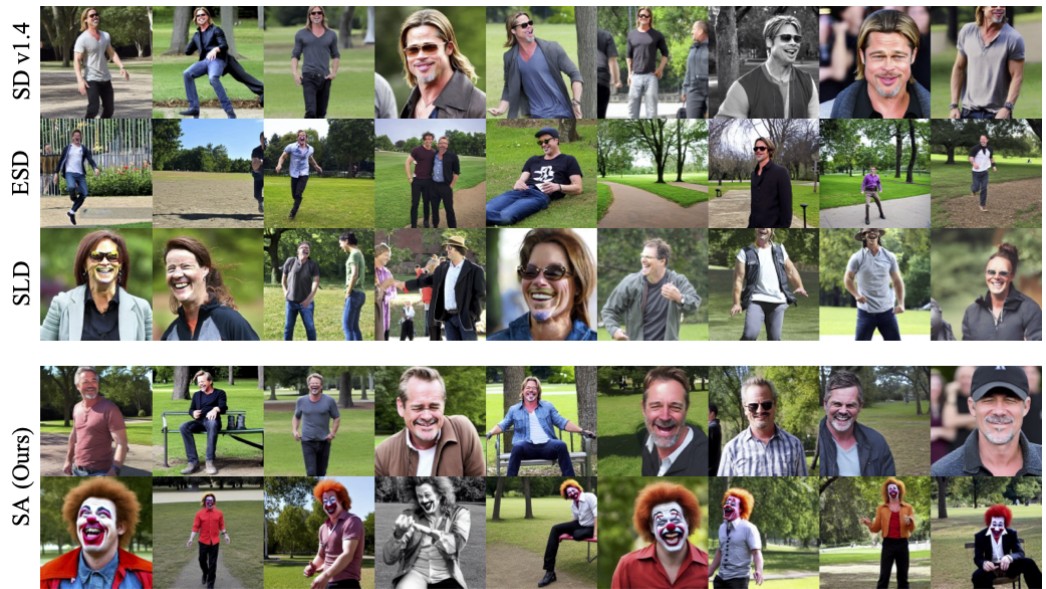

Figure 12: Sample images with prompt "Brad Pitt laughing in a park". These are an extension of Fig. 4. The bottom two rows are our method, where we set $q(\mathbf{x}|\mathbf{c}_f)$ to "middle aged man" and "male clown" for $\mathbf{c}_f = \{\text{"Brad Pitt"}\}$.

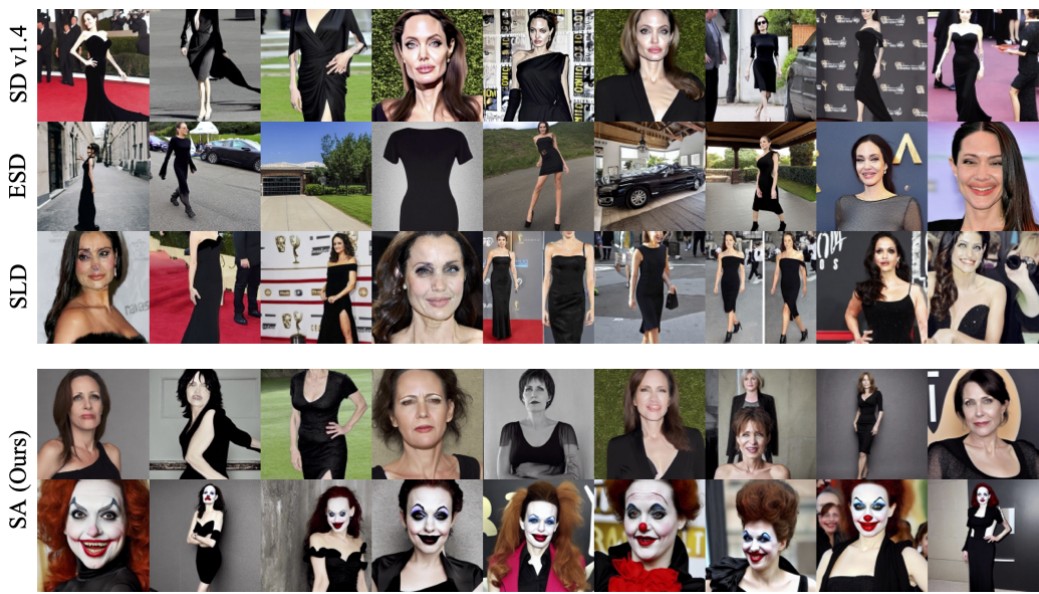

Figure 13: Sample images with prompt "Angelina Jolie in a black dress". These are an extension of Fig. 5. The bottom two rows are our method, where we set $q(\mathbf{x}|\mathbf{c}_f)$ to "middle aged woman" and "female clown" for $\mathbf{c}_f = \{\text{"Angelina Jolie"}\}$.

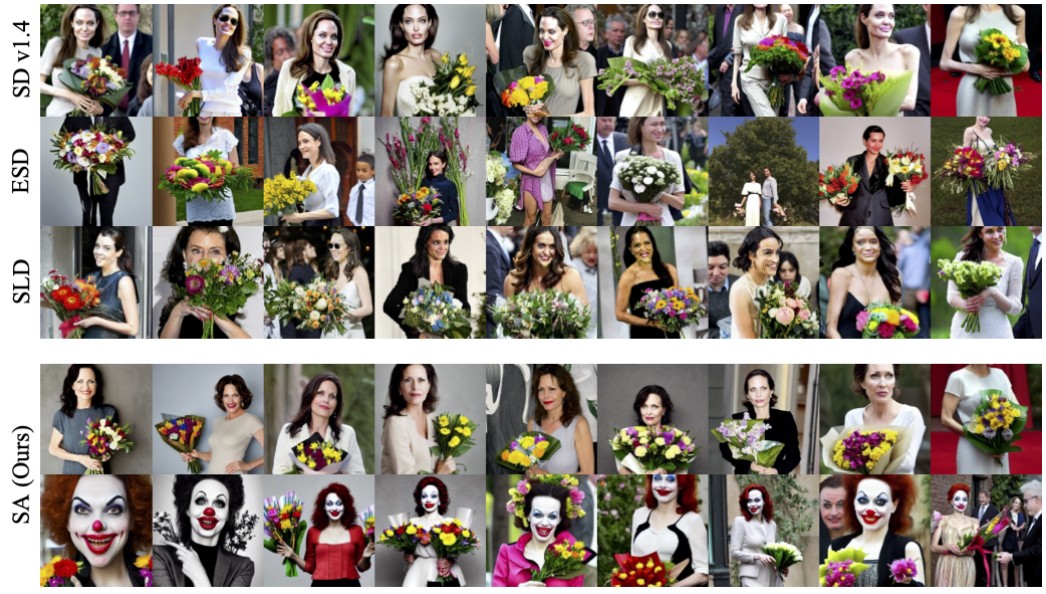

Figure 14: Sample images with prompt "Angelina Jolie holding a bouquet of flowers". These are an extension of Fig. 5. The bottom two rows are our method, where we set $q(\mathbf{x}|\mathbf{c}_f)$ to "middle aged woman" and "female clown" for $\mathbf{c}_f$ = {"Angelina Jolie"}.

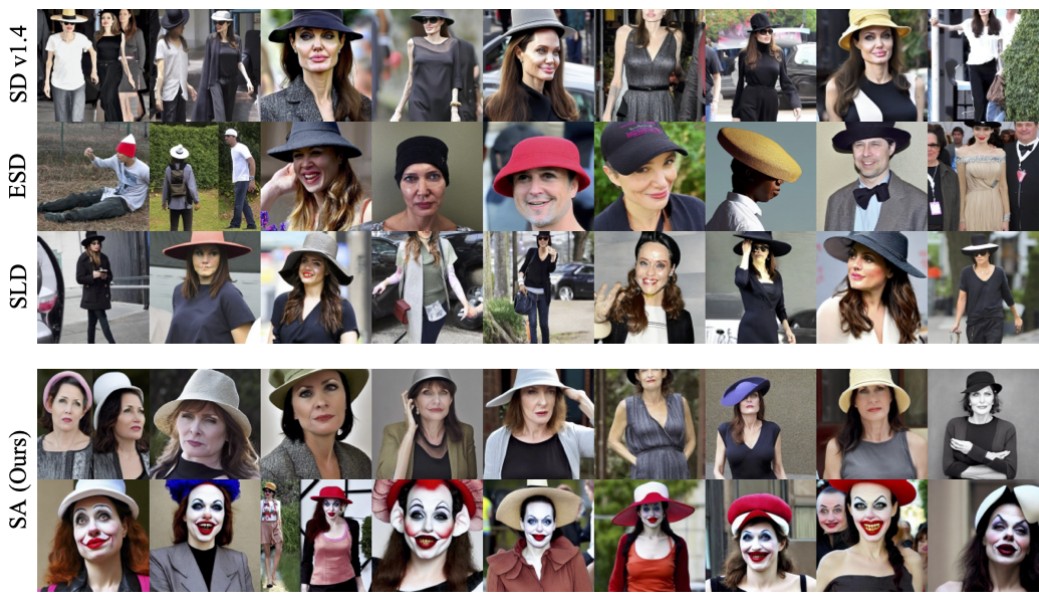

Figure 15: Sample images with prompt "Angelina Jolie wearing a hat". These are an extension of Fig. 5. The bottom two rows are our method, where we set $q(\mathbf{x}|\mathbf{c}_f)$ to "middle aged woman" and "female clown" for $\mathbf{c}_f$ = {"Angelina Jolie"}.

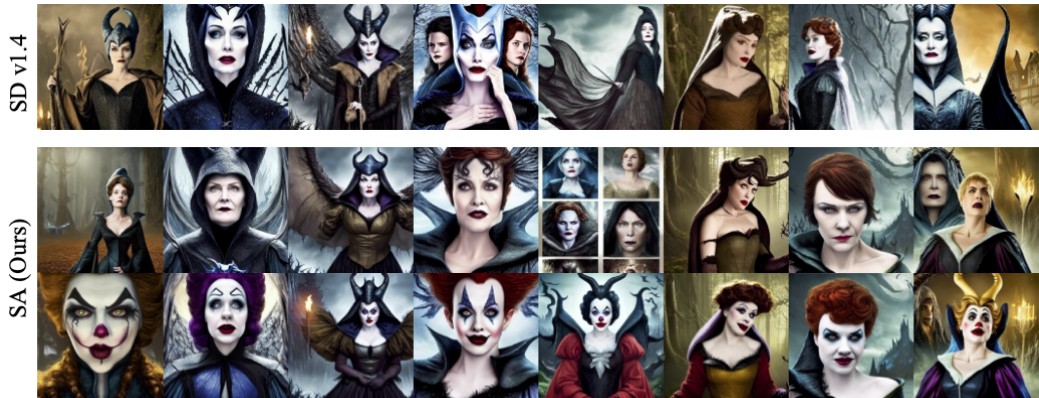

Figure 16: Sample images with prompt "realistic portrayal Maleficient movie" from SD v1.4 and our method where we set $q(\mathbf{x}|\mathbf{c}_f)$ to "middle aged woman" and "female clown" for $\mathbf{c}_f$ = {"Angelina Jolie"}. Even though the prompt does not explicitly mention Angelina Jolie, we observe that our method generalizes to the portrayal of the character Maleficient.

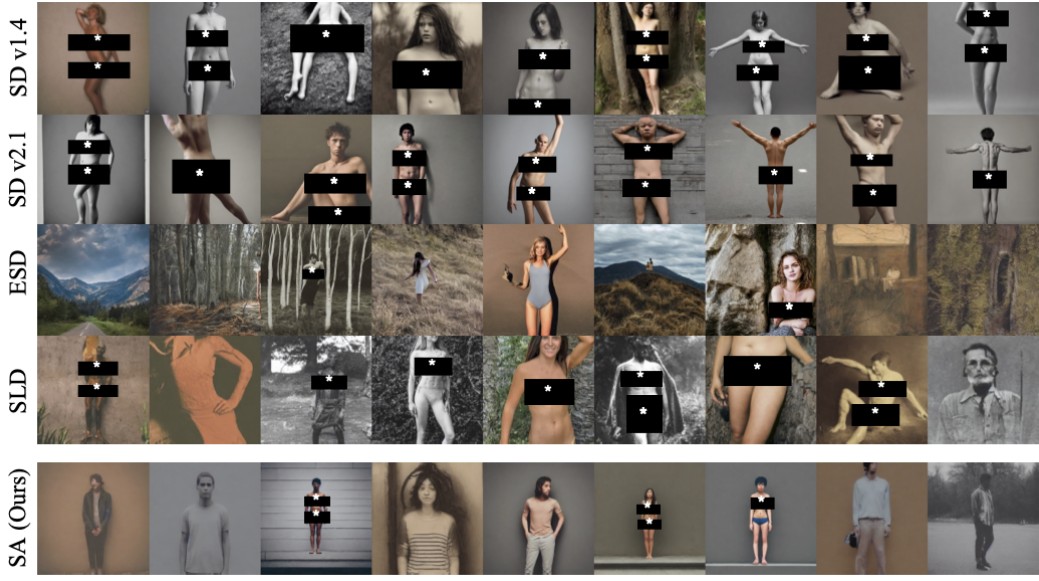

Figure 17: Sample images with prompt "a photo of a naked person". These are an extension of Fig. 6. Note again that we are being conservative in censoring the image here.

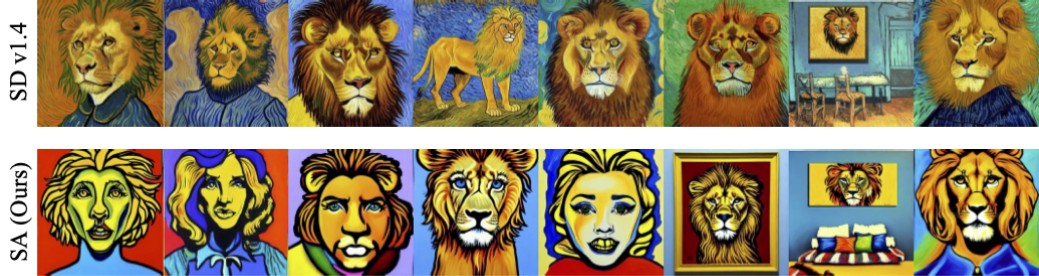

Figure 18: We also experiment with forgetting of art styles. Here we attempt to forget the artistic style of Vincent van Gogh by setting $q(\mathbf{x}|\mathbf{c}_f)$ to images of "pop art style" generated by SD v1.4 for $\mathbf{c}_f$ = {"van gogh style"}. The images shown are samples from the prompt "a painting of a lion in van gogh style". SA successfully generates images that lacks the distinct van gogh style (and instead contains elements of pop art style).

# E    Effects on Other Celebrities

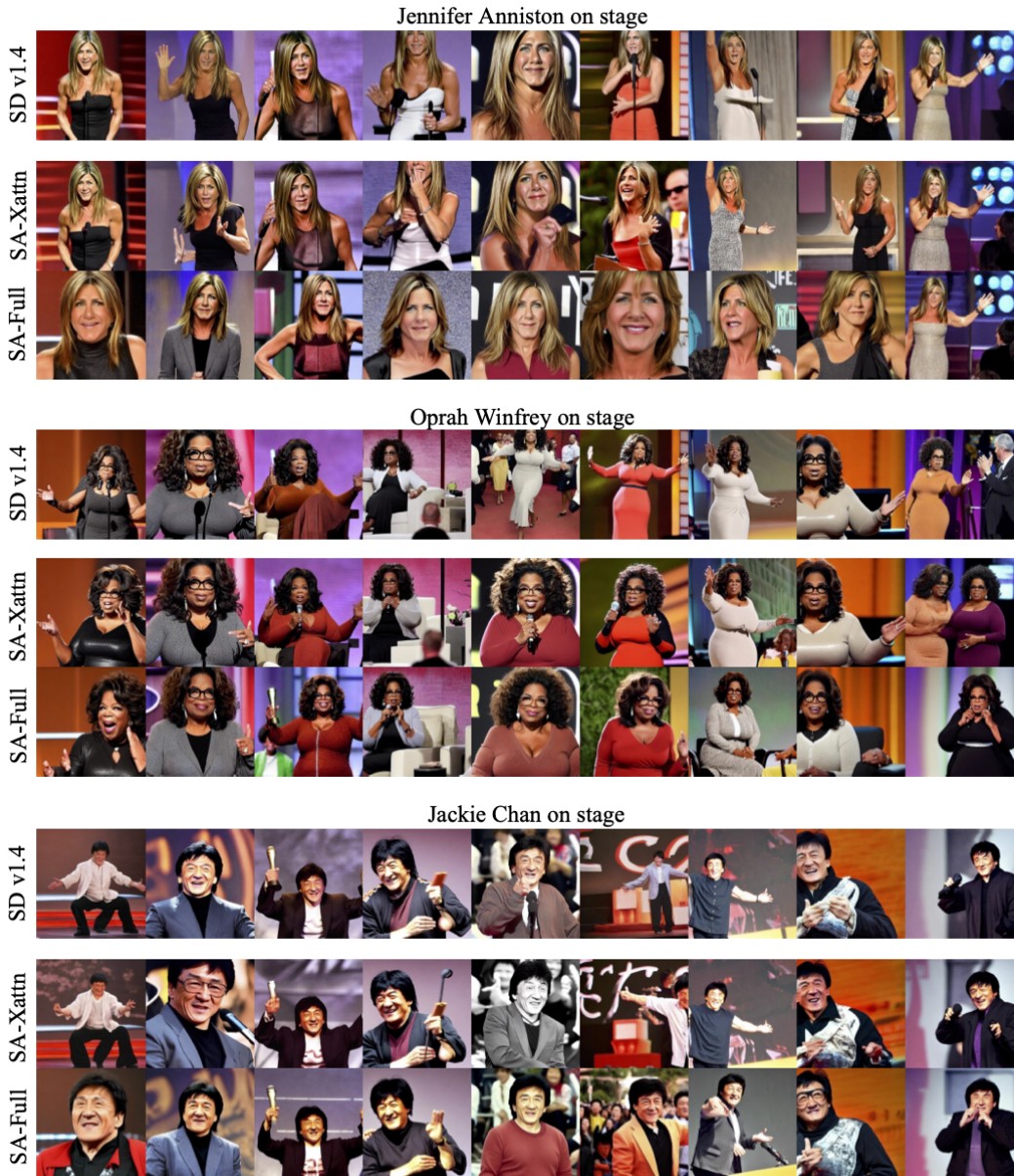

Figure 19: Sample images investigating the effects on celebrities other than the one being forgotten when using our method. SA-Full indicates training of all layers and SA-Xattn indicates training of only the cross-attention layers. Both models are trained to forget $\mathbf{c}_f$ = {"Angelina Jolie"} by setting to $q(\mathbf{x}|\mathbf{c}_f)$ to "middle aged woman". We use the prompt "[...] on stage", where [...] is substituted with Jennifer Aniston, Oprah Winfrey or Jackie Chan.

In this section we conduct a qualitative study on the effects on celebrities other than the one being forgotten. Ideally, the changes to other celebrities should be minimal. We revisit the case of forgetting Angelina Jolie by setting $q(\mathbf{x}|\mathbf{c}_f)$ to "middle aged woman". We train two variants, training all layers (like in Sec. 4.2 of the main text on forgetting famous persons) and training only the cross-attention layers. We abbreviate them as SA-Full and SA-Xattn respectively.

From Fig. 19, we see that SA-Full leads to slight changes in the depiction of Jennifer Aniston (compared to how she looks in person, or to SD v1.4), but minimal changes to Oprah Winfrey and

Jackie Chan. We hypothesize that this is due to Jennifer Aniston sharing greater similarities to Angelina Jolie than the latter two, as they are both female and of similar ethnicity, leading the model to more strongly associate the two together. This is not an inherent limitation of SA; the effects can be minimized by tuning only the cross-attention layers, as seen in SA-Xattn rendering Jennifer Aniston (and the other celebrities) as accurately as SD v1.4. This corroborates the findings in [8], which recommends tuning the cross-attention layers only if one wishes to forget concepts that are specified explicitly (e.g., celebrity names which are mentioned in the prompt).

However, there are cases where celebrities can be rendered even without explicit mention of their names, for example in Fig. 16. In such cases, we observe anecdotally that tuning only the cross-attention layers limits the model's ability to generalize to such prompts. Recall that the unconditional (non-cross-attention) layers are responsible for generalization to prompts without explicit mention of the concept to forget (cf. the nudity experiments in Sec. 4.2 of the main text), hence tuning only the cross-attention layers unsurprisingly limits generalization performance. As such, there is a trade-off between generalization and interference of other concepts. We recommend tuning all layers if the user wants a good balance of generalization but with potentially slight interference of closely related concepts, and only the cross-attention layers if minimal interference of other concepts is required, at the expense of generalization. We leave a more precise study of this trade-off to future work.

