# OpenReview forum: "Selective Amnesia: A Continual Learning Approach to Forgetting in Deep Generative Models"
_NeurIPS.cc/2023/Conference — NeurIPS 2023 spotlight_

### Official Review · Reviewer_ehLH · 2023-06-11

**Soundness:** 3 good
**Presentation:** 3 good
**Contribution:** 3 good
**Rating:** 6
**Confidence:** 4

**Summary:**

This paper draws inspirations from continual learning and proposes a Selective Amnesia method to selectively forget concepts in pre-trained generative models. The proposed method is implemented in a widely-used framework of Bayesian continual learning, together with generative replay to optimize the objective. Experiments on several datasets demonstrate the effectiveness of the proposed method.

**Strengths:**

1. The paper is well-organized and easy to follow.
2. The reverse application of the continual learning process to forget certain concepts is an interesting idea.
3. The experimental results, especially for the case study on Stable Diffusion, seem to be remarkable. This may contribute to realistic applications in erasing harmful concepts and protecting privacy.

**Weaknesses:**

1. The idea of selective or graceful forgetting has been explored in continual learning. In particular, AFEC [1] also considers a similar Bayesian-based framework and controls a forgetting rate to incorporates a non-informative prior $p(\theta)$. Interestingly, I find Eq.(2) also derive the prior $p(\theta)$. Therefore, a comparison of the proposed Selective Amnesia and AFEC is necessary.
2. The experiments only compare Selective Amnesia and Original. Although the results of selective forgetting seem to be remarkable, is it possible to compare with representative baselines of continual learning and/or machine unlearning?
3. A primary motivation of this work is to avoid the computational overhead of retraining all old data. However, the proposed method relies on strong generative replay, which potentially deviates from the motivation. I encourage the authors to compare the computational overhead of generative replay, and also consider it when comparing with other baselines.

[1] AFEC: Active Forgetting of Negative Transfer in Continual Learning. NeurIPS 2021.

**Questions:**

My major concerns include the comparison of AFEC, more advanced baselines and computational cost of generative replay.  Please refer to the weakness.

**Limitations:**

The authors have discussed the limitations and potential negative impact.

---

> ### Author Rebuttal · Authors · 2023-08-08
>
> We thank the reviewer for acknowledging the quality, strong experimental results and interesting approach of our work. Below we provide our responses to the concerns raised by the reviewer.
>
> > Therefore, a comparison of the proposed Selective Amnesia and AFEC is necessary.
>
> Thank you for bringing AFEC to our attention. We will include a discussion of AFEC in the revised manuscript. The major difference between AFEC and our work SA is that AFEC tackles the traditional setting of sequential tasks learning. AFEC proposes an adaptive posterior over prior tasks when learning a new task in a traditional continual learning setting, which allows conflicting weights from old tasks to be forgotten if they will negatively affect the new tasks. In our work, we would like complete forgetting of $D_f$. Applied to AFEC, this means setting $\beta=1$ in Eq 6, i.e., completely ignoring the posterior over old tasks. This reduces to relying solely on catastrophic forgetting of $D_f$, which we tried in early experiments and found did not lead to sufficient forgetting of $D_f$ due to strong conditioning of conditional generative models. Regarding the prior term in Eq 6, $p(\theta)$ appears from applying Bayes rule over the posterior of the task to remember. This resembles the prior in Eq 2 of our work, which arises from Bayes rule over $D_f$. Similar to us, it seems that the prior term in AFEC is left out of the optimization.
>
> > ...is it possible to compare with representative baselines of continual learning and/or machine unlearning.
>
> We were unable to find suitable continual learning baselines as continual learning is concerned with *preventing* catastrophic forgetting, while we are interested in promoting forgetting.  As mentioned above, relying solely on catastrophic forgetting of $D_f$ led to suboptimal results.
>
> We also could not find suitable machine unlearning baselines for our specific setting of forgetting classes in conditional variational generative models. Based on a recent machine unlearning review [1], the methods covered were applied mainly to discriminative models. To our knowledge, there are no machine unlearning works targeted at generative models. Next, we considered several representative methods in machine unlearning and sought to investigate if they could be extended to generative models.
>
> As discussed in Sec 2.3, several prior works had limiting assumptions that prevented them from being applied to our setting. For instance, [2] requires altering the original training process of the model via dataset sharding. [3] requires a variational posterior over the original model weights, whereas we only have a MLE point estimate. [4] assumes that the weights of the model that has forgotten $D_f$ only differ from the original model by a small Gaussian error, and proposes to directly modify the weights of the network. As the method is demonstrated only on classification, it is unclear if the assumption holds in the generative case. Regardless, we ran preliminary experiments of [4] on the MNIST VAE, with the simplified Fisher forgetting method of Sec 4 of the paper (as SA also utilizes the FIM). We found this to be unstable due to the inversion of the FIM ($F^{-1/4}$ term).
>
> We also considered [5], which we will include in Sec 2.3 of the revised manuscript. [5] proposes to tune a model to remove the influence of certain datapoints but requires the gradients during the original training process to be cached, which we do not have.
> In general, it appears that methods in the machine unlearning literature [1] often have limiting assumptions and are more suitable for discriminative settings. In considering other avenues for baselines, we also considered other methods designed for generative models (which were pointed out by reviewer 3). However, we found that these methods will not work in our specific setting of conditional variational models, or were not appropriate for forgetting. Extending these methods to our setting would constitute major separate research. For additional details regarding these methods, please refer to our response to reviewer 3.
>
> > the proposed method relies on strong generative replay, which potentially deviates from the motivation.I encourage the authors to compare the computational overhead of generative replay, and also consider it when comparing with other baselines.
>
> We thank the reviewer for the suggestion on the overhead of GR, which we will include in our revised manuscript. The cost of GR comes in generating $D_r$. In the case of SD, we use a representative set of 5000 images generated from prompts by ChatGPT, which takes approximately 6 hours on a single GPU. As these prompts are general, we reuse the same $D_r$ across all SD experiments. The GR overhead is far smaller than the computational resources needed to retrain SD from scratch (reported to require 256 A100s for 150,000 GPU hours). Also, the cost of generating $D_r$ can be amortized over many runs, making it negligible compared to the cost of training SA or baselines like ESD (which we discuss further in our response to reviewer 3).
>
> Thank you again for your review and feedback. We hope our response has addressed your concerns. If so, we hope you will consider updating your review and score.
>
> References
>
> [1] Nguyen, T., et al. "A survey of machine unlearning."
>
> [2] Bourtoule, L., et al. "Machine unlearning."
>
> [3] Nguyen, Q.P., Low, B.K.H. and Jaillet, P.. "Variational bayesian unlearning."
>
> [4] Golatkar, A., Achille A,, and Soatto, S.. "Eternal sunshine of the spotless net: Selective forgetting in deep networks."
>
> [5] Wu, Y., Dobriban E., and Davidson S.. "Deltagrad: Rapid retraining of machine learning models."

---

> > ### Comment · Reviewer_ehLH · 2023-08-15
> > **Reviewer response**
> >
> > Thank you for replying to my comments and questions. This rebuttal has addressed my concerns. I appreciate the idea of introducing graceful forgetting in generative models, which is importance for realistic applications. The use of ChatGPT for generative replay is also an innovative idea for this technical route (especially for Diffusion Models). I increase my score and prefer to accept this paper.

---

> > > ### Author Response · Authors · 2023-08-15
> > >
> > > Thank you very much for your feedback and we appreciate that you have raised your score!

---

### Official Review · Reviewer_VhMr · 2023-07-01

**Soundness:** 2 fair
**Presentation:** 3 good
**Contribution:** 3 good
**Rating:** 6
**Confidence:** 3

**Summary:**

The paper aims to forget selected concepts from deep generative models trained with variational inference with a focus on the trustworthiness such as malicious prompts. The paper introduces a continue learning approach that combines EWC and GR to maximize the likelihood (or ELBO) on the remembered set. Experimentally, the paper evaluates forgetting qualities for simple models on standard datasets and for large scale stable diffusion on real world datasets.

**Strengths:**

The forgetting task is very meaningful and important as we are now facing such trustworthy challenges in the large model era.

The continual learning approach that combines EWC and GR is a good idea and solves several practical difficulties as discussed in the methodology section.

There is theory that could guarantee improved likelihood after using the surrogate loss.

The framework applies to several types of models trained with likelihood or ELBO.

The results on forgetting identity and malicious contents from stable diffusion are interesting and potentially useful.

**Weaknesses:**

The selection of $q$ is not well studied in experiments and seems arbitrary. There should be more detailed explanation on the selection of $q$.

The paper lacks analysis on the computational resources needed for the continual learning approach.

The experimental results for simple models on standard datasets are not compared to baseline methods. There are a few methods for similar goals in the wild including data redaction, feature unlearning, model rewriting, model taming, etc.

The experimental results for stable diffusion do not seem to be better than baseline methods, especially the "erasing concepts from diffusion models".

Some minor points:

I recommend the authors to discuss more related research (mentioned above) in the related work section so that readers can have a better sense of the literature.

Please check typos and unbalanced parentheses in the math equations.

**Questions:**

What is the criteria of selecting $q$? Why does the selection of $q$ leads to mapping of concepts (line 262)?

How much computational resource (memory and time) do you need to continual-learn stable diffusion? Do you fine-tune a subset of layers or train all layers?

What is the advantage of the proposed method compared to baseline methods for stable diffusion?

**Limitations:**

The paper addressed limitations in the last section.

---

> ### Author Rebuttal · Authors · 2023-08-08
>
> We thank the reviewer for acknowledging the motivations and usefulness of our work. Below we provide our responses to the reviewer's concerns.
>
> > ...detailed explanation on the selection of q
>
> The choice of $q$ can be understood theoretically from Corollary 1-- a greater difference between $q$ and the distribution over $D_f$ means a lower likelihood over $D_f$. In general, we want $q$ to be far from $D_f$. For instance, in the experiments of Table 1, $q$ is chosen to be uniform noise as it is intuitively far from the distribution of natural images. However, we stress that Corollary 1 serves as a heuristic, and users are free to choose $q$ that is most relevant for their use case. In our SD experiments, we attempt to forget certain celebrities and a semantically-relevant choice is to have the model generate unrecognizable persons.
>
> > Why does the selection of q leads to mapping of concepts?
>
> SA trains the model to forget $D_f$ by instead generating concepts from $q$ when conditioned on $c_f$. For instance, in the experiments in Sec 4.2, as we chose $q$ to represent images of middle-aged men, the model generates unrecognizable men when prompted for Brad Pitt.
>
> > Do you fine-tune a subset of layers or train all layers?
>
> Our method does not make assumptions about network architecture, thus we tune all layers except in the nudity experiments, where we tune only unconditional layers of SD. One could also tune only cross-attention layers to minimize interference with other concepts (explored in appendix E).
>
> > ...analysis on the computational resources needed
>
> Appendix B mentions computational resources which we will expand to include more details. In brief, we used 2 RTX A5000s for DDPM experiments and 4 A6000s for SD experiments. Training with SA takes 4-5 hours on DDPM and 20 hours on SD. Memory usage in SD is approximately 40GB. Note that we did not optimize for computational efficiency in our reported experiments; by tuning hyperparameters in preliminary experiments, we could achieve similar performance with 2 GPUs and 6 hours of training. For comparison, the ESD baseline also uses 2 GPUs and around 2 hours of training. We believe additional performance gains can be achieved with further tuning.
>
> > ...results for simple models on standard datasets are not compared to baseline methods… including data redaction, feature unlearning, model rewriting, model taming, etc *and* ...discuss more related research (mentioned above) in the related work section
>
> Thank you for the suggested keywords. We surveyed representative works but the methods are not applicable to forgetting concepts in conditional variational generative models, or require major extensions. We elaborate below, and will include these discussions in the revised manuscript:
>
> [1] and [2] pertain to GANs and Normalizing Flows, respectively. The former requires the discriminator as feedback for the generator for data redaction. [4] employs exact likelihood computation of NFs to reduce the likelihood over $D_f$. Variational models lack access to exact likelihoods, and Sec 3.2 shows that minimizing the ELBO over $D_f$ leads to poor results.
>
> [3] implicitly assumes that the generator’s latent space has disentangled features, which does not apply to our conditional models. For instance, a given latent $z$ can generate all ten digits of MNIST, $x_i=G_\theta(z,c_i), i=0, …, 9$, by changing the conditioning signal. Hence, it is unclear how one would apply [3] to conditional models.
>
> [4] is work on image editing, such as removing watermarks from images, by directly modifying a single layer’s weights in a generator. The paper's experiments focused only on GANs and required finding the best layer to tune for specific applications. This method was not designed for forgetting concepts in conditional variational models and we are unsure whether it qualifies as a suitable baseline. Preliminary experiments with the provided code (on GANs) on more drastic changes that forgetting necessitates did not yield desired results, e.g, altering entire roofs of churches to domes led to severe visual artifacts.
>
> > The experimental results for stable diffusion do not seem to be better than baseline methods, especially the "erasing concepts from diffusion models"
>
> We do not claim that our method is strictly better than the strong baselines, at least purely in terms of the classifier metrics. We emphasize that the metrics only provide a partial view of the results; the compared methods have qualitatively different behaviors, as discussed in Sec. 4.2 and App. C. ESD (and SLD) steers the model in arbitrary, uncontrollable directions away from the concept, which results in generated images lacking semantic relevance. For instance, ESD frequently produces images without faces or persons, such as houses or cars when prompted for Angelina Jolie (Fig 13 of app.), or mountains and forests in the nudity experiments (Fig 5). This is reflected in the high proportion of generated images without faces (over 30%) in Table 2 of the appendix. By choosing $q$ to be images of unrecognizable persons, SA produces fewer images without faces (about 5%). Compared to SLD, SLD tends to produce distorted faces with visual artifacts. We believe these aspects should be considered when comparing methods.
>
> > Please check typos and unbalanced parentheses in the math equations.
>
> We thank the reviewer for pointing out the typos. We have corrected the unbalanced parentheses in Thm 1 and Corollary 1.
>
> [1] Kong, Z., and K. Chaudhuri. "Data redaction from pre-trained gans."
>
> [2] Malnick, S., Shai A., and Ohad F.. "Taming a Generative Model."
>
> [3] Moon, S., S. Cho, and D. Kim. "Feature unlearning for generative models via implicit feedback."
>
> [4]  Bau, D., et al. "Rewriting a deep generative model."
>
> We hope that we have addressed your concerns. If so, we kindly request to consider updating your review and score.

---

> > ### Comment · Reviewer_VhMr · 2023-08-18
> > **Reply**
> >
> > Thanks the authors for the reply.
> >
> > The response addressed some of my questions.
> >
> > Regarding baseline models, please include the discussion in the related work section.
> >
> > I like the idea of more "semantic relevance" of the proposed model. Table 2 is good. However, I still feel there lack a systematic evaluation if you are trying to show this is a general phenomenon that does not restrict to faces or some specific examples.
> >
> > I'm increasing my score to 6.

---

> > > ### Author Response · Authors · 2023-08-21
> > >
> > > We thank the reviewer for raising their score and for providing additional feedback. We agree with the reviewer that a systematic evaluation of forgetting a variety of concept types is desirable. However, this would require a general framework (vis-a-vis our classifier-based approach), such as evaluation of the model’s likelihoods, and benchmark datasets which is an interesting an avenue for future research. Within the context of this paper, we have conducted additional experiments on forgetting artist styles (for e.g., forgetting “van gogh style” by setting $q$ to images of “pop art style”) and obtained positive results. We will include these in the revised appendix. Taken together with the main quantitative and qualitative results, we believe this shows that Selective Amnesia works on a variety of concepts.

---

### Official Review · Reviewer_n35v · 2023-07-05

**Soundness:** 3 good
**Presentation:** 4 excellent
**Contribution:** 3 good
**Rating:** 7
**Confidence:** 4

**Summary:**

In this work, authors introduce Selective Amnesia - a new method for selective forgetting of particular concepts in generative modeling without access to the training data. Authors propose to use Continual Learning methods - namely, Elastic Weights Consolidation and Generative Replay to retrain the base model with carefully selected additional data that promote forgetting of examples conditioned on particular conditional values without affecting the other ones. In particular, to enforce forgetting, authors introduce a surrogate objective that, instead of just minimizing the probability of generating examples from a distribution that needs to be forgotten, substitutes it with a different one e.g., random noise.


**Strengths:**

- This submission tackles an important problem crucial for the reliable use of recent ML applications.
- The main contribution is clear, seems to be well thought and is well presented both in terms of intuition and details.
- The proposed evaluation is extensive, convincing, and well-structured. The experiment results conducted on several datasets with multiple methods are impressive. Simple experiments were implemented with Variational Autoencoders, while the more extensive ones with high-quality datasets employed diffusion models.
- The work is well-written and easy to follow.
- Additionally to the main contribution, this work introduces an interesting reformulation of the EWC method.

**Weaknesses:**

The proposed method does not really enforce forgetting the particular concept but rather promotes replacing it with a different one. This might lead to fake associations between conditional values and generated outputs. For example, as presented in this work, forgetting “Brad Pitt” generations by learning to generate a “male clown” instead might also affect other generations with prompts similar to “Brad Pitt” - e.g. other actors. This problem should be more evident when trying to forget more general concepts by replacing them with different ones. To make the method more general, it would be interesting to find a technique that automatically finds the suitable surrogate dataset.

Small:
- Small detail: "most prior work on continual learning for generative models are applied to GANs [19, 20], while our work is primarily concerned with variational models." - This is not true, see, for example (1,2,3,4,5). Those methods tackle the problem of continual learning of variational models, so the fact that they already exist does not affect the novelty of this work. However, I think it would be beneficial to mention at least [1], that first introduced EWC for Variational Autoencoders. I don’t believe the rest should be cited, but I just wanted to point those to authors as potentially interesting for any future works.



1. Nguyen, Cuong V., et al. "Variational Continual Learning." International Conference on Learning Representations.
2. Egorov, Evgenii, Anna Kuzina, and Evgeny Burnaev. "BooVAE: Boosting approach for continual learning of VAE." Advances in Neural Information Processing Systems 34 (2021): 17889-17901.
3. Achille, Alessandro, et al. "Life-long disentangled representation learning with cross-domain latent homologies." Advances in Neural Information Processing Systems 31 (2018).
4. Mundt, M., Pliushch, I., Majumder, S., Hong, Y., & Ramesh, V. (2022). Unified probabilistic deep continual learning through generative replay and open set recognition. Journal of Imaging, 8(4), 93.
5. Deja, Kamil, et al. "Multiband VAE: Latent Space Alignment for Knowledge Consolidation in Continual Learning." IJCAI 2022Limitation - how precise is the forgetting presented in this method? What happens if the concept we want to forget is highly correlated with another one? E.g. if we take CelebA dataset, I think it might be very hard to forget the “wearing lipstick” class without forgetting “heavy makeup” one. From a more precise point of view, I believe this question relates to the problem of how adequate is the approximation of p(x|c)p_f(c) with generations with concepts that we want to forgot/remember?
What would happen if we simply skip the EWC regularisation term and continue model retraining with only two ELBOs with generative replay?

**Questions:**

- How precise is the forgetting presented in this method? What happens if the concept we want to forget is highly correlated with another one? E.g. if we take CelebA dataset, I think it might be very hard to forget the “wearing lipstick” class without forgetting “heavy makeup” one. From a more precise point of view, I believe this question relates to the problem of how adequate is the approximation of p(x|c)p_f(c) with generations with concepts that we want to forgot/remember?

- What would happen if we simply skip the EWC regularisation term and continue model retraining with only two ELBOs with generative replay?

**Limitations:**

Authors address the main limitations of the work except from the one described in weaknesses/questions

---

> ### Author Rebuttal · Authors · 2023-08-08
>
> We would like to thank the reviewer for acknowledging the strong motivation and contributions of our work, as well as our extensive experimental results. Below we provide our responses to the concerns raised by the reviewer.
>
> > What happens if the concept we want to forget is highly correlated with another one?
>
> We thank the reviewer for raising this issue of “leakage” to correlated concepts. We have documented this effect in appendix E, where we show that forgetting “Angelina Jolie” leads to slight changes when generating “Jennifer Anniston”. We discussed a way to mitigate this issue, which is to tune the cross-attention layers only in Stable Diffusion. We view this “leakage” issue as a potential double-edged sword that could be beneficial in situations where we would like to forget highly correlated concepts. For instance, in the nudity experiments in Sec 4.2, one would prefer the remapping of clothed persons to generalize to related prompts, such as when the prompt includes artist styles that typically contain nudity.
>
> > ...it would be interesting to find a technique that automatically finds the suitable surrogate dataset
>
> This is an interesting idea. In this work, our approach was to enable the user to pick $q$ as the surrogate distribution as its choice may be context-dependent (e.g., due to cultural differences). That said, the system can suggest potential $q$'s that are commonly accepted to be benign yet retain the generation capabilities of the model. We will add this discussion to the revised manuscript as future work.
>
> > ...see, for example (1,2,3,4,5)... I think it would be beneficial to mention at least [1], that first introduced EWC for Variational Autoencoders
>
> Thank you for pointing out these related works and we will revise our discussion in Sec 2.2. As the reviewer notes, [1] addresses the continual learning problem and was applied to a VAE to sequentially learn MNIST classes.  The method uses variational approximation to calculate the posterior over the parameters of a model for the T-th learning task, given parameters for the T-1th learning task.  Our setting here is different in that we work on forgetting specific concepts rather than continual learning; we will amend our related works section to discuss the relationship.
>
> > What would happen if we simply skip the EWC regularisation term and continue model retraining with only two ELBOs with generative replay?
>
> As suggested by the reviewer, we ran our method on DDPM to forget the airplanes class in CIFAR10 by turning off EWC completely. This is achieved by setting $\lambda=0$. We kept all other hyperparameters identical to experiments in Table 1. After training, we evaluated the image quality on the remaining 9 classes. We obtained an FID of 45.7, a precision of 0.078 and recall of 0.803, which suggests that image fidelity significantly deteriorated without the EWC term. This result is overall in line with the ablations in Table 1, which show that image fidelity decreases as we decrease the strength of the EWC term.
>
> [1] Nguyen, Cuong V., et al. "Variational continual learning." arXiv preprint arXiv:1710.10628 (2017).
>
> Thank you for your positive review and we hope that we have addressed your remaining concerns. If there are any further issues, please let us know.

---

> > ### Comment · Reviewer_n35v · 2023-08-14
> > **Rebuttal comment**
> >
> > Thank you for addressing all of my comments and questions and for pointing out the discussion on the concept leakage in the appendix. I agree that this might be a double-edged sword, although this is a significant limitation of this work if someone wants to apply it in practice. Maybe the careful selection of the surrogate distribution could help in preventing changes in the correlated concepts?
> >
> > I am satisfied with the reply, and I am strongly convinced that this work should be accepted for NeurIPS.

---

> > > ### Author Response · Authors · 2023-08-15
> > >
> > > Thank you for your positive remarks! Regarding the selection of $q$ to further prevent concept leakage, this may be possible in conjunction with other schemes, e.g., training only the cross-attention in Stable Diffusion. Another approach might be to automatically remap highly-correlated concepts to related $q$'s, but this remains future work.

---

### Official Review · Reviewer_bzv7 · 2023-07-06

**Soundness:** 3 good
**Presentation:** 3 good
**Contribution:** 3 good
**Rating:** 6
**Confidence:** 3

**Summary:**

This paper presents a technique called Selective Amnesia, which enables controllable forgetting of concepts in pretrained deep generative models. The authors show that this technique can be applied to a variety of models, including text-to-image and VAEs, and can be framed from the perspective of continual learning. The paper discusses two popular approaches used in their work: Elastic Weight Consolidation and Generative Replay. The authors argue that this technique can be used to prevent the generation of harmful, misleading, and inappropriate content by deep generative models.

**Strengths:**

1. The manuscript ventures to address a vital and emergent challenge — ensuring content safety within the ambit of generative artificial intelligence. This issue is of critical importance, given the increasing prevalence and capabilities of generative models, coupled with the potential risks and implications associated with their misuse. The authors' focus on such a topical concern is commendable and lends significant relevance and timeliness to their work.

2. The authors introduce a model inspired by continual learning — a concept that, while established in the broader field of machine learning, presents a novel approach in the specific context of generative model safety. This inventive application of continual learning principles adds a fresh dimension to the discourse on generative model safety, thereby enhancing the manuscript's contribution to this field of study.

3. The selection of datasets for the empirical evaluation is impressively diverse. The authors have conducted experiments on a broad spectrum of datasets, ranging from simpler, well-established ones such as MNIST to more complex, real-life image datasets. This range serves to validate the robustness and versatility of their model across different levels of complexity and in varied contexts, adding significant credibility to their findings. Furthermore, this choice reflects a meticulous approach to experimental design, allowing the model's performance to be tested and evaluated under a wide array of conditions.

4. The quality of exposition is commendable. The paper presents a clear narrative, beginning with the intuitive premise, followed by a comprehensive description of the model, including a highly useful algorithmic representation. The logical sequencing of these sections facilitates reader comprehension. The authors have succeeded in presenting a complex topic in an accessible and cogent manner.

**Weaknesses:**

The manuscript is of high quality and is presented competently, with no major flaws readily apparent. However, my primary concern pertains predominantly to the scope and rigor of the quantitative evaluation.

1. The current quantitative evaluation is exclusively conducted on relatively simple datasets. While this serves as a valid starting point, extending the quantitative evaluation to more challenging datasets would provide a more comprehensive perspective on the model's performance. Although this might present additional difficulties, I would encourage the authors to devise some form of quantitative measure, even if it requires human assessment for accuracy. Such a quantitative comparison would help to mitigate potential selection bias and lend greater credence to the results.

2. A more thorough discussion on the chosen baseline models and the current state-of-the-art models would be greatly beneficial. By elucidating how these models relate to the proposed model, and where they diverge, the authors could provide a clearer context for their work. Such discourse would enhance the readers' understanding of the field's landscape and allow them to appreciate the distinctiveness and the value of the authors' contribution more deeply.

3. The comparative analysis currently conducted is primarily qualitative in nature. However, this leaves a gap in the evaluation, as quantitative comparisons can provide unique insights that are not always captured by qualitative analysis. Quantitative comparisons can offer more objective, replicable, and measurable findings, which are highly valuable in establishing the proposed model's superiority over existing models.

Addressing these concerns would add depth to the study, providing a more rigorous and holistic evaluation of the proposed model. Such improvements would be of significant value, helping to fortify the authors' claims and making a more compelling case for the paper's contributions to the field.

**Questions:**

Please refer to the above sections for detailed discussion.

**Limitations:**

Yes, there is a section at the end of the paper discussing limitations and future research opportunities.

---

> ### Author Rebuttal · Authors · 2023-08-08
>
> We would like to thank the reviewer for acknowledging the strong motivations, novelty and quality of our work. Below we provide our responses to the concerns raised by the reviewer.
>
> > ...extending the quantitative evaluation to more challenging datasets would provide a more comprehensive perspective on the model's performance *and* ..even if it requires human assessment for accuracy
>
> Quantitative results for the Stable Diffusion experiments are shown in Tables 2, 3 and 4 in Appendix C (due to space constraints). We will make this clearer in the main text.
>
> In brief, we evaluated these experiments with well-established pretrained classifiers: the GIPHY Celebrity Detector (GCD) and NudeNet nudity classifier; the latter was employed in prior works on SLD and ESD. We agree with the reviewer that additional quantitative measures, such as human evaluations, would provide a more holistic evaluation due to the many factors one can consider when forgetting concepts in Stable Diffusion. For instance, we have noted in Sec 4.2 of the paper that our model produces more semantically-relevant images (far fewer faceless images), and the faces that our model generates are of higher quality (particularly compared to SLD), despite having a higher GCD score (where lower is better) than ESD and SLD.  We will add to Sec. 5 that conducting human evaluations is part of our future work.
>
> > A more thorough discussion on the chosen baseline models and the current state-of-the-art models would be greatly beneficial.
>
> We included discussion and comparisons with works from the machine unlearning literature as well as baselines in Stable Diffusion (ESD and SLD) in Sec. 2.3 and 2.4, respectively. Discussions on qualitative and quantitative results from experiments between our method and ESD and SLD are also included in Sec 4.2. We will include additional discussions in the related works section on feature unlearning, data redaction and model rewriting in deep generative models. We believe this expanded discussion will be reasonably comprehensive given space constraints. If the reviewer has specific suggestions for relevant works or comparisons, please let us know.
>
> We hope that we have addressed your concerns. If so, we kindly request to consider raising your score.

---

> > ### Comment · Reviewer_bzv7 · 2023-08-21
> >
> > Thanks for the rebuttal. It has addressed most of my conners.
> > With everything considered, I'd raise my rating to 6.

---

> > > ### Author Response · Authors · 2023-08-21
> > >
> > > We thank the reviewer for their positive feedback and for raising their score!

---

### Author Rebuttal · Authors · 2023-08-08

Thank you to the reviewers for their comments and feedback. We appreciate the positive comments and that the reviewers find our continual learning approach to concept forgetting to be meaningful and technically interesting. Based on the comments, we have revised the paper to include (i) additional comparisons to related work pointed out by the reviewers, (ii) details on computational cost, and (iii) text to clearly point out relevant portions of the Appendix (e.g., on concept leakage and additional quantitative results). Please see below for detailed responses to each reviewer.

---

### Decision · Program_Chairs · 2023-09-21

**Decision:**

Accept (spotlight)

**Comment:**

From the reviews, rebuttal and discussion, there was unanimous agreement that the problem being explored and technical contributions of the paper meet the standards for NeurIPS and further should be of broad interest to the community.